# ADMN: A Layer-Wise Adaptive Multimodal Network for Dynamic Input Noise and Compute Resources

**Jason Wu**
Electrical and Computer Engineering
University of California, Los Angeles
jaysunwu@g.ucla.edu

**Yuyang Yuan**
Electrical and Computer Engineering
University of California, Los Angeles
yuanyuyang@g.ucla.edu

**Kang Yang**
Electrical and Computer Engineering
University of California, Los Angeles
kyang73@g.ucla.edu

**Lance Kaplan**
DEVCOM Army Research Laboratory
US Army
lance.m.kaplan.civ@army.mil

**Mani Srivastava**[*]
Electrical and Computer Engineering
University of California, Los Angeles
mbs@ucla.edu

## Abstract

Multimodal deep learning systems are deployed in dynamic scenarios due to the robustness afforded by multiple sensing modalities. Nevertheless, they struggle with varying compute resource availability (due to multi-tenancy, device heterogeneity, etc.) and fluctuating quality of inputs (from sensor feed corruption, environmental noise, etc.). Statically provisioned multimodal systems cannot adapt when compute resources change over time, while existing dynamic networks struggle with strict compute budgets. Additionally, both systems often neglect the impact of variations in modality quality. Consequently, modalities suffering substantial corruption may needlessly consume resources better allocated towards other modalities. We propose ADMN, a layer-wise **A**daptive **D**epth **M**ultimodal **N**etwork capable of tackling both challenges - it adjusts the total number of active layers across all modalities to meet strict compute resource constraints, and continually reallocates layers across input modalities according to their modality quality. Our evaluations showcase ADMN can match the accuracy of state-of-the-art networks while reducing up to 75% of their floating-point operations.

## 1 Introduction

**Background:** Multimodal deep learning systems fusing sensory data from various modalities are the standard for accurate, robust sensing [1, 2]. Accordingly, these multimodal systems are invaluable in highly dynamic environments, where a given input modality's quality-of-information (QoI) can vary drastically across samples. Low QoI data is corrupted, noisy, or otherwise degraded in a manner reducing its informativeness. Fluctuations in a modality's QoI can occur slowly (e.g., lighting conditions over the day) or rapidly (e.g., battlefield settings or unstable sensor feeds).

---

[*]Mani Srivastava holds concurrent appointments as a Professor of ECE and CS (joint) at the University of California, Los Angeles, and as an Amazon Scholar at Amazon. This paper describes work performed at UCLA and is not associated with Amazon.

39th Conference on Neural Information Processing Systems (NeurIPS 2025).

**Challenges:** Although multimodal deep learning systems are generally robust to variable QoI, a key challenge surrounds the *computational efficiency*. Most state-of-the-art multimodal networks employ *static provisioning* in which multimodal inputs are processed by a fixed architecture [3, 4] regardless of their individual utility. Consequently, valuable resources may be wasted on low-QoI modalities. Recent work has explored dynamic networks that train policy networks to reduce computation for *easy samples* [5, 6, 7, 8]. Unfortunately, explicit consideration of highly dynamic QoI variations among modalities has been largely overlooked. *We hypothesize that flexibly allocating computational resources among modalities in accordance with each modality's QoI on a per-sample basis can substantially improve model performance in compute-limited settings.*

Additionally, current works also neglect the challenge of *dynamic compute resource availability*. The environments where multimodal systems are most relevant often impose *temporally variable but strictly bounded* compute budgets. The maximum budget varies with time according to factors such as thermal throttling, energy fluctuations, or multi-tenancy, and cannot be exceeded at any given moment. Neither statically provisioned models nor dynamic networks are equipped to function under such constraints. Most dynamic networks optimize for *average-case efficiency*, without mechanisms to constrain the worst-case compute beyond the total cost of the full network. The only partially compatible works are those performing *model selection with gating networks* [6, 8, 9], which can be adapted to varying compute resource availability by training a set of models for each budget. Aside from requiring an unreasonable amount of training resources, it also impedes the standard practice of loading pretrained weights prior to finetuning [10], as there do not exist publicly available pretrained weights for every possible compute budget. *We hypothesize that a single network initialized with pretrained weights, which can dynamically adjust its resource usage, offers an effective solution to the challenge of fluctuating compute resources.*

**Proposed Solution:** We propose ADMN, an **A**daptive **D**epth **M**ultimodal **N**etwork jointly tackling the challenges of adaptation to both dynamic compute resources and variable QoI inputs. While these challenges are agnostic to the multimodal fusion method (e.g., data-level [11], embedding-level [12], and late [13]), we focus specifically on embedding-level fusion due to applicability and ease of implementation. Figure 1 provides a high-level depiction of ADMN. Following the standard for embedding-level fusion, each modality is processed with an independent backbone before undergoing fusion with a transformer encoder.

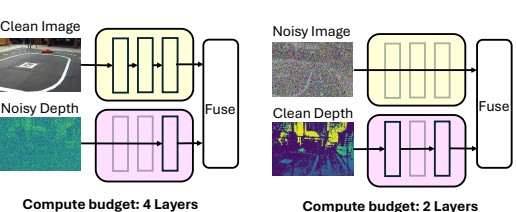

Figure 1: Overview of ADMN. Variable depth backbones adapt to both changing compute resources and input noise characteristics

First, ADMN addresses the challenge of *dynamic compute resources* through adaptive backbones containing adjustable layer configurations. With the same set of model weights, ADMN activates a subset of backbone layers according to the available compute resources. We ensure compatibility with pretrained weight initializations by injecting the LayerDrop technique [14] into both the backbone pretraining and multimodal network finetuning processes. This requires the adaptation of the unimodal, text-only LayerDrop technique to not only Vision Transformers, but also multimodal networks. Such a strategy produces a novel multimodal network whose backbones are resilient to missing layers at test-time, allowing for the usage of fewer layers during resource scarcity.

Second, given an established layer budget, ADMN addresses the challenge of *dynamic QoI* by adapting the choice of selected backbone layers according to each modality's QoI. ADMN learns a multimodal controller on top of the adaptive backbones to perform layer allocation. Unlike existing works, we explicitly structure the controller around dynamic QoI by introducing *corruption-aware supervision* into the training loss. However, as it is reliant on the presence of labeled corruption information during training, we propose an alternative *autoencoder-based initialization* (ADMN_AE) to emphasize the QoI without corruption labels. The controller is trained end-to-end for a particular layer budget through Gumbel-Softmax Sampling [15] and the straight-through estimator [16].

We benchmark ADMN against several baselines to demonstrate its ability to preserve accuracy while simultaneously minimizing energy and latency costs. ADMN is tested on both multimodal localization

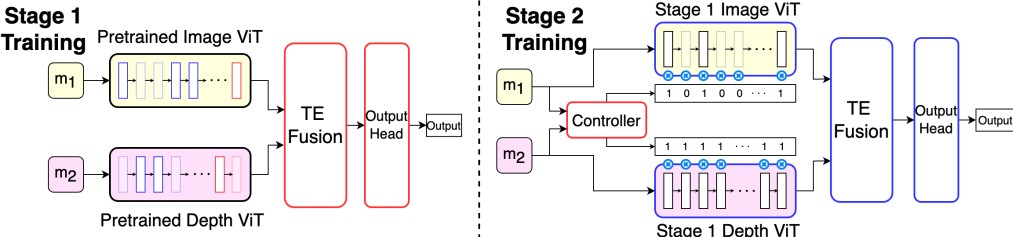

Figure 2: `ADMN` architecture. [Gray box]: dropped layer, [Blue box]: frozen layer, [Red box]: tunable layer. TE: Transformer Encoder.

and classification tasks with realistic sensor corruptions representing dynamic QoI, reinforcing its generality and applicability. `ADMN` can match the performance of larger state-of-the-art models while reducing FLOPS by up to 75% and latency by up to 60%. We release our code at `https://github.com/nesl/ADMN`. Our contributions are summarized as below:

- We present the first layer-wise adaptive multimodal network where resource allocation among modality backbones is dictated by QoI characteristics and temporally variable but strictly bounded compute budgets at inference time for every sample. We evaluate `ADMN` with realistic sensor corruptions modeling dynamic QoI.

- We create a general framework for training layer-adaptive multimodal networks by injecting the LayerDrop technique into both unimodal backbone pretraining and multimodal finetuning, and systematically evaluate it across various modality combinations and datasets.

- We design a multimodal perceptual controller that explicitly attends to modality QoI through either *corruption-aware supervision* when training-time corruption labels are provided, or *autoencoder-based initialization* when they are not.

- We introduce a low-complexity method of training a multimodal controller that meets strictly bounded compute budgets instead of reducing average-case computation.

## 2 Related Work

**Early Exiting in Unimodal Networks.** Early Exiting has been explored extensively in unimodal networks to improve the *average-case inference efficiency* [17, 18, 5]. Methods like DeeBERT [17] and PABEE [18] use confidence thresholds to halt computation for simpler inputs. However, extending such techniques to dynamic multimodal QoI and strictly bounded, variable compute budgets is nontrivial. Specifically, they neglect to consider the impact of low-QoI samples even in the *unimodal* setting, while `ADMN` addresses the complex interplay of dynamic *multimodal* QoI. Moreover, Early Exiting is unable to constrain the *worst-case* inference computation beyond the total network cost, clashing with the need to accommodate strictly bounded compute budgets.

**Unimodal Sub-Networks.** Instead of performing input-aware network adaptation, other works prune a single large network at runtime to accommodate varying computational resources [19, 14]. *Once-For-All* [19] proposed an algorithm pruning a large network across depth, width, kernel size, and resolution, showcasing competitive performance against state-of-the-art neural architectural search baselines. *LayerDrop* [14] introduced a layer-wise dropout rate during pretraining for on-demand layer reduction at test time. We leverage LayerDrop as the foundation for `ADMN`, but emphasize that `ADMN`'s contributions can easily be applied to other inference-time pruning techniques.

**Dynamic Inference for Multimodal Systems.** To address the inefficiency of static multimodal networks, dynamic networks [8, 6, 20, 7, 21] leverage input-dependent forward paths. DynMM [8] and DynaFuse [9] train a set of expert networks with different modalities and perform network selection at inference time. AdaMML [6] and Listen to Look [20] leverage multimodal information to eliminate temporal redundancy in videos. ACF [7] dynamically replaces certain modules with lightweight networks according to the input for greater efficiency. Despite these advances, existing methods (1) overlook significant QoI variations arising from corruption, (2) utilize or discard entire modalities without fine-grained control, and (3) fail to consider *fixed* resource budgets. `ADMN` addresses

these gaps by introducing two methods of performing layer allocation with an explicit focus on relative modality QoI, while also accounting for a strictly bounded compute budget.

# 3 Methodology

## 3.1 Problem Description

In real-world multimodal systems, sensor data suffers from *dynamic QoI variations*. Environmental factors such as inclement weather and lowlight can corrupt the utility of a modality, while sensor-specific factors such as camera ISO or audio gain also play a factor. Additionally, environmental QoI corruptions may unequally affect modalities. For example, fog may affect vision (heavily) and depth (partially), but not audio. Furthermore, we also impose a strictly bounded, time-varying compute constraint to emulate factors such as thermal throttling or variable available energy. Thus, the goal of ADMN is to perform correct modality resource allocation according to the diverse sensor QoI variations, all while ensuring that the total allocated resources adhere to a strict budget.

A typical embedding-level fusion multimodal network is illustrated on the left side of Figure 2. It extracts unimodal embeddings with modality-specific backbones, condenses them into one joint embedding via self-attention with a *Transformer encoder*, and obtains an output from the joint embedding. ADMN builds upon this architecture by proposing two novel advancements. First, we introduce a *layer-wise adaptive multimodal network* comprised of dynamic modality backbones, thus enabling inference-time adaptation to any layer budget. Second, ADMN proposes a QoI-aware controller that dynamically allocates the layer budget optimally among modality backbones (i.e., transformer layers), allowing it to greatly outperform static models with the same layer budget.

## 3.2 ADMN Architecture

Figure 2 shows the task-agnostic architecture of ADMN, which involves a two-stage training process.

**Stage 1: LayerDrop Finetuning.** We first initialize each unimodal backbone with a set of general weights (e.g., ImageNet weights) pretrained with LayerDrop [14]. Subsequently, the multimodal network is finetuned with LayerDrop on the desired task while freezing the earlier backbone layers (shown in blue) to prevent overfitting. Stage 1's objective is to adapt the pretrained weights to the specific target task, while also training the later Fusion and Output layers to accommodate diverse embeddings arising from various backbone layer configurations.

**Stage 2: Controller Training.** We freeze all the Stage 1 network weights and train a controller with either *corruption-aware supervision or autoencoder initialization* to allocate a fixed budget of $L$ layers in accordance to each sample's relative modality QoI. From the multimodal inputs, the controller outputs a discrete sequence summing to $L$ representing the selection of backbone layers.

## 3.3 Stage 1: LayerDrop Finetuning

We adapt LayerDrop [14], originally designed for *unimodal text transformers*, to support arbitrary multimodal transformers through integration into unimodal pretraining and multimodal finetuning.

### 3.3.1 Vanilla LayerDrop

LayerDrop [14] enabled on-demand test-time depth reduction of textual transformers by training with a layer-wise dropout rate. By stochastically dropping out the transformer layers, the network is trained to function with only a subset of layers. For an inference-time layer budget $L$, they introduced an "every-other" dropout strategy where layers are dropped in an alternating fashion starting from the middle. We refer readers to the original work for greater detail.

### 3.3.2 Multimodal LayerDrop

ADMN extends LayerDrop to Vision Transformer-style networks (ViT) [22] by first integrating them into the unimodal backbone pretraining process. We pretrain 12-Layer visual and audio backbones on the ImageNet-1k [23] and AudioSet [24] datasets, respectively. Rather than performing supervised training, which suffers from convergence issues, we pretrain both backbones with the Masked

Autoencoder (MAE) approach [25, 26]. We enforce a LayerDrop rate of 0.2 in each layer of the ViT encoder while fixing the decoder. By employing LayerDrop in MAE pretraining, the model learns to reason in the presence of missing layers.

Subsequently, the MAE pretrained weights are loaded into the appropriate backbones of a multimodal (e.g., vision-depth) neural network to be finetuned on a specific corrupted dataset. As shown in Figure 2, the majority of the early backbone layers are frozen during finetuning, with later layers adjusting to the new task. A LayerDrop rate of 0.2 is maintained during the finetuning process in all the backbones, allowing the remaining learnable layers (Fusion and Output) to adapt to the countless combinations of missing layers. This process ultimately creates a task-specific multimodal network supporting adaptive backbone layer configurations at inference time.

**Full-Backbone LayerDrop.** One unique challenge with *multimodal* LayerDrop is the need to subject individual backbones to extreme dropout conditions. While dropping all layers in a *unimodal* network is rare, one may frequently drop all layers of a heavily corrupted modality in a *multimodal* network. Unfortunately, the typical LayerDrop training ratio of 0.2 in a standard 12 layer ViT is unlikely to expose the network to such a case during training. To remedy this, we employ full-backbone dropout during training time, establishing a 10% chance that all layers of a given modality's backbone will be dropped out independently of the 0.2 LayerDrop rate.

### 3.4 Stage 2: Controller Training

The controller selectively activates backbone layers in the frozen Stage 1 network according to the relative modality QoI and total layer budget $L$ on a per-sample basis to minimize task loss, as shown in Figure 2. Ideally, the controller should also remain lightweight in operations, latency, and size.

#### 3.4.1 Controller Architecture

In contrast to previous works [21, 5, 8] which train controllers to reduce computation in *the average case with no regard to variable QoI or fixed resource budgets*, ADMN's controller is explicitly constructed around both factors. Figure 3 reveals the structure of the controller. First, we downsample the inputs and pass them through modality specific lightweight convolutional networks. These convolutions aim to produce embeddings containing information solely regarding each modality's QoI, which is sufficient from low-resolution data. With $M$ modalities, the convolutional networks produce $M$ total noise embeddings, which are fused by a transformer encoder into one embedding $e_{\text{corr}}$ representing the QoI of every input modality. From $e_{\text{corr}}$, we can obtain a set of raw logits

$$\pi = \text{MLP}_\pi(e_{\text{corr}}) \text{ where } \pi \in \mathbb{R}^C, \ C = \sum_{i=1}^{M} |b_i| \tag{1}$$

where $|b_i|$ is the total number of backbone layers for modality $m_i$. In essence, $\pi$ represents an allocation of $L$ available layers among $C$ total backbone layers, with values dependent on the QoI characteristics of every input sample.

#### 3.4.2 QoI-Aware Training

Ideally, the convolutional layers will automatically focus on each modality's QoI to provide correct layer allocations from the task-specific loss $\mathcal{L}_{model}$. Nevertheless, as we later show in Section 4.4, supervision with purely the task loss is insufficient, where the controller fails to properly attend to QoI and outputs unsuitable layer allocations (Section A.7). Thus, we introduce two methods enabling QoI-awareness: *corruption-aware supervision* when the training dataset contains QoI labels, and *autoencoder-based initialization* (termed as `ADMN_AE`) when they do not.

**Corruption-Aware Supervision.** We supervise the controller with an additional corruption loss $\mathcal{L}_{corr}$. If the corruption is quantifiable (e.g., camera ISO, Gaussian Noise), we predict a corruption vector $\hat{\sigma}_m = \text{MLP}_{\text{corr}}(e_{\text{corr}})$ containing the corruption values for all modalities. The corruption loss is the MSE loss $\mathcal{L}_{corr} = \sum_{i=1}^{M} |\hat{\sigma}_{m_i} - \sigma_{m_i}|$, where $\sigma$ represents the ground truth corruption value. If the corruption is difficult to quantify but can be grouped into $K$ categories (e.g., weather annotations), we obtain a set of logits $f \in \mathbb{R}^K$, where $f = \text{MLP}_{\text{corr}}(e_{\text{corr}})$, and set $\mathcal{L}_{corr}$ as the cross-entropy loss between $f$ and the category labels. We optimize over the joint loss $\mathcal{L}_{total} = \mathcal{L}_{model} + \mathcal{L}_{corr}$, which explicitly instructs the model to attend to the modality QoI.

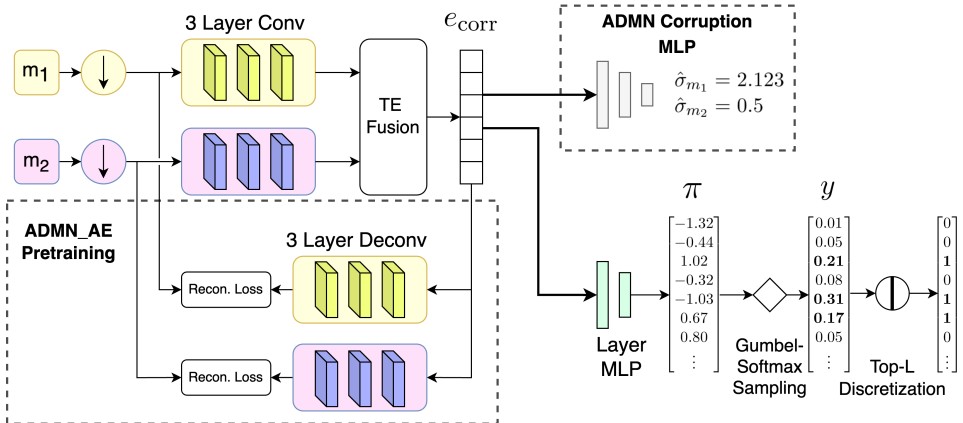

Figure 3: Detailed depiction of the ADMN controller.

**Autoencoder-Based Initialization.** In many scenarios, there is a lack of any ground truth QoI information. The key intuition of the *autoencoder-based initialization* approach is that the compression and reconstruction objectives of autoencoders result in a well-organized latent space that group similar samples together [27, 28]. Since the QoI corruptions are significant enough to impact downstream accuracy, they are vital to the reconstruction objective. Consequently, an autoencoder trained on variable QoI data is likely to structure the latent space in a manner that reflects each sample's QoI properties. Figure 3 illustrates the structure of ADMN_AE's autoencoder pretraining. The *encoder* is comprised of the controller's perceptual components (i.e., convolution layers and fusion transformer), while the *decoder* performs reconstruction from $e_{\text{corr}}$ with modality specific deconvolution layers. With autoencoder pretraining, the controller's perceptual layers learn to attend to input modality QoI without the need for any QoI labels. We then load the encoder weights into the controller, freeze them to retain QoI understanding, and learn the layer MLP allocations with the singular loss $\mathcal{L}_{model}$.

### 3.4.3 Differentiable Layer Selection

Although the logits $\pi$ provide information on which layers to select, activating a given layer is a binary decision. Such categorical decisions interrupt the flow of gradients and prevent end-to-end training. Traditionally, unimodal early-exit networks [5] employ Gumbel-Softmax Sampling [15] to approximate a categorical distribution while retaining differentiability. These networks employ decision networks *at every layer* of the model and leverage Gumbel-Softmax Sampling to decide whether to execute that particular layer. Coupled with a resource-aware loss, these techniques can train dynamic networks that reduce computation in the *average case on easy samples*. However, ADMN substantially differs from previous dynamic networks in that ADMN predicts (1) the entire allocation of layers at the *beginning* of model execution that (2) sum to a hard layer budget $L$. Simply applying Gumbel-Softmax Sampling to ADMN is insufficient, as the approach cannot select $L$ total layers among all modality backbones.

We devise a method to select $L$ total layers in a differentiable manner. Initially, we perform standard Gumbel-Softmax sampling with a temperature of 1 to obtain $y$. This allows multiple high-value logits to be represented in the resulting probability distribution, better accommodating the selection of $L$ layers. This is accomplished at the expense of the highly desirable near-discrete behavior of the low-temperature Gumbel-Softmax. We introduce a subsequent *top-L discretization* on $y$ to activate only $L$ layers, and maintain gradient flow through the *straight-through estimator* [16]. The downstream layers receive the discretized value, while the gradients received are copied over to the continuous logits $y$, allowing for gradient flow. We provide additional justification in Appendix A.2.

Table 1: GDTM Localization Error (cm) (↓) with 6-16 Layers of Budget

| | GDTM Dataset | | | | | | | | | | | |
| | Gaussian | | | | Lowlight | | | | Blur | | | |
| Method | 6 | 8 | 12 | 16 | 6 | 8 | 12 | 16 | 6 | 8 | 12 | 16 |
| --- | --- | --- | --- | --- | --- | --- | --- | --- | --- | --- | --- | --- |
| Upper Bound | 29.6 | | | | 18.8 | | | | 9.4 | | | |
| ADMN | 51.4 | 39.0 | **33.1** | 30.3 | 49.5 | 23.9 | **18.0** | 17.3 | **11.8** | **11.2** | **9.8** | 9.9 |
| ADMN_AE | 53.6 | 38.4 | 33.5 | 29.4 | **35.3** | 22.8 | 18.7 | 17.6 | 14.0 | 11.4 | 9.8 | **9.6** |
| MNS | 39.3 | **32.1** | 57.5 | 73.0 | 117.5 | 83.9 | 117.6 | 116.8 | 109.5 | 114.2 | 117.7 | 74.0 |
| Naive Alloc. | 112.5 | 97.6 | 46.9 | 31.0 | 90.3 | 67.3 | 27.1 | 17.7 | 81.4 | 50.6 | 12.5 | 9.8 |
| Naive Scratch | **39.1** | 36.7 | 40.4 | 37.3 | 117.5 | 84.6 | 117.6 | 117.5 | 117.8 | 116.2 | 117.7 | 93.6 |
| Image Only | 67.5 | 53.2 | 53.7 | — | 56.8 | 45.6 | 46.2 | — | 15.4 | 11.2 | 10.3 | — |
| Depth Only | 85.0 | 61.7 | 58.6 | — | 72.1 | 64.9 | 63.4 | — | 25.7 | 22.6 | 22.3 | — |

Table 2: MMFI Classification Accuracy (↑)

| | MMFI (Gaussian) | | | |
| Method | 6 | 8 | 12 | 16 |
| --- | --- | --- | --- | --- |
| Upper Bound | 44.44% | | | |
| ADMN | 35.03% | **39.25%** | 41.92% | 43.31% |
| ADMN_AE | **36.16%** | 38.43% | **42.18%** | **44.40%** |
| MNS | 3.70% | 3.70% | 3.70% | 3.70% |
| Naive Alloc. | 5.56% | 12.96% | 29.01% | 42.90% |
| Naive Scratch | 3.70% | 3.70% | 3.70% | 3.70% |
| Image Only | 18.83% | 18.52% | 23.15% | — |
| Depth Only | 17.59% | 30.86% | 32.72% | — |

Table 3: AVE Classification Accuracy (↑)

| | AVE (Rain + Background Noise) | | | |
| Method | 6 | 8 | 12 | 16 |
| --- | --- | --- | --- | --- |
| Upper Bound | 71.19% | | | |
| ADMN | **57.07%** | **62.95%** | **67.48%** | 66.60% |
| ADMN_AE | 52.95% | 62.30% | 66.77% | 66.67% |
| MNS | 25.81% | 26.56% | 17.21% | 13.34% |
| Naive Alloc. | 36.16% | 46.89% | 65.71% | **67.95%** |
| Naive Scratch | 23.94% | 26.56% | 18.70% | 13.22% |
| Image Only | 47.76% | 54.11% | 57.85% | — |
| Audio Only | 36.41% | 40.52% | 42.64% | — |

# 4 Evaluations

## 4.1 Experimental Setup

### 4.1.1 Datasets and Corruptions

The GDTM localization dataset [12] involves distributed nodes each containing modalities RGB, depth, mmWave radar, and multichannel audio. The target is a remote-controlled car on an indoor track. The MM-Fi [29] human activity recognition dataset contains 40 subjects, 27 total activities, and modalities RGB, depth, mmWave, and WiFi. We focus on the visual modalities (RGB, depth) for these two datasets. Finally, we evaluate on the audiovisual AVE [30] dataset with 4143 videos covering 28 classes, from which we use both RGB and audio modalities. These datasets do not naturally contain varying modality QoI, so we synthetically corrupt the modalities on a per-sample basis (Section A.7, Section A.3) to simulate realistic sensor corruptions as follows:

**Gaussian Noise**: We add $N(0, \sigma_{i_j})$ to each modality $i$'s input data. Each modality defines a set of $N_i$ standard deviations $\{\sigma_{i_1}, \sigma_{i_2}, ...\sigma_{i_{N_i}}\}$ from which $\sigma_{i_j}$ is drawn for each sample. This setting can represent systems with unstable links injecting different levels of noise, or sensors with different settings such as camera ISO levels. We apply this to the *RGB and depth modalities of the GDTM and MM-Fi datasets with $N_i = 4$.*

**Rain:** For the *outdoor RGB samples of the AVE Dataset*, we also explore simulating rainfall and haze corruption, following the technique in [31].

**Lowlight:** We mimic lowlight corruption through a combination of gamma correction, color shift, and additive noise [32]. We create moderately and severely impaired lowlight *RGB samples from GDTM* while leaving the IR depth unchanged. However, under normal lighting (RGB unchanged), we emulate a light-saturated IR depth sensor by utilizing a frame of sunlight saturated IR depth. We also add lowlight corruption to both *rainy outdoor and clean indoor RGB samples of the AVE Dataset.*

**Blur:** We employ Gaussian Kernel blurring with two kernel sizes to emulate different severity of blur on *RGB images from the GDTM.*

**Background Noise:** We mix in wind audio for outdoor events and sound from a standing fan for indoor events to corrupt the *AVE dataset's audio modality.*

#### 4.1.2 Baselines

**Upper Bound:** The full 12 layers are allocated to each backbone (no dropout), representing the maximum layer budget.

**Naive Allocation:** Given a layer budget $L$ and $M$ modalities, we naively allocate $\frac{L}{M}$ layers to each backbone following "every-other" allocation.

**Image/Depth/Audio Only:** All $L$ layers are allocated to one modality, valid only for $L \leq 12$

**Naive Scratch:** We train a *new network from scratch without LayerDrop for every layer budget $L$* on the downstream task with each backbone containing $\frac{L}{M}$ layers.

**Modality Network Selection (MNS):** Most existing dynamic networks ignore fixed resource budgets and cannot be directly compared to `ADMN`. We adapt the network selection technique from [6, 8, 9], which train a set of "expert models" and use a gating network to select among them to create the *MNS* baseline. This technique is also frequently referred to as "mixture-of-experts". *MNS* trains an individual unimodal network for each modality and one multimodal network where *each model is trained from scratch to conform to a specific layer budget*. In contrast, ADMN requires only the training of one single backbone network with LayerDrop, scaling much more effectively with the number of budgets, while also reaping the advantage of pretrained weight initializations.

Essentially, *Upper Bound*, *Naive Allocation*, and *Image/Depth/Audio Only* are *allocation baselines* operating on the same finetuned Stage 1 backbone, while *Naive Scratch* and *MNS* train new networks.

### 4.2 Main Results

**GDTM Localization Error.** Table 1 highlights that both `ADMN` (corruption supervised) and `ADMN_AE` (autoencoder pretrained) drastically outperform all the allocation baselines regardless of the corruption, especially at small layer budgets. Under Lowlight with 8 layers of budget, `ADMN` and `ADMN_AE` localize within 5 cm of the upper bound, while the next best baseline incurs almost 27 cm of localization error. The performance difference is the smallest under Blur, which we attribute to the strength of the visual backbone. While the heavily blurred RGB images are uninformative to the human eye (Section A.7), the visual backbone can still produce accurate localizations, and the controller correctly prioritizes image for layer allocation despite heavy blur. This exemplifies a strength of `ADMN` – learning allocations from the task loss prevents errors from preconceived human bias. In comparison to Naive Scratch and MNS, `ADMN` handily outperforms them with the exception of 6 and 8 layer Gaussian corruption. Due to the need to train networks from scratch (cannot leverage pretrained weights for arbitrary layer budgets), these two baselines struggle to converge with deeper networks and more complex corruptions. `ADMN_AE`'s great performance also demonstrates that knowledge of the ground truth corruption is unnecessary.

**Classification Accuracy.** Tables 2 and 3 depict the classification accuracy for the MM-Fi and AVE Datasets, respectively, from 6 to 16 Layers. We observe similar trends as the GDTM dataset, but notice weaker results for the Naive Scratch and MNS baselines. These classification tasks with changing environments and temporal dependencies are considerably more complex than the single-car GDTM localization task. Consequently, the accuracy of networks trained from scratch suffers. This serves to illustrate the importance of our LayerDrop training process – it enables the adaptation of pretrained networks to any arbitrary layer budget.

**Latency and FLOPs.** We show the meaningful reduction in latency (ms) and Giga-Floating Point Operations (GFLOPs) of `ADMN` in Figure 4. Although the controller accounts for a significant proportion of the latency at smaller layer budgets ($\sim 20\%$), the high throughput at these latencies ($> 150$ fps for GDTM) surpasses most sensor sampling rates. Moreover, `ADMN`'s controller utilizes a negligible amount of FLOPs. The controller constitutes about 1% and 0.2% of the model's total operations for GDTM and MM-Fi, respectively, at *the fewest allocation of 6 layers*. When viewing Tables 1, 2, and 3 in context of these metrics, we can observe the significance of `ADMN`. For instance, under Blur corruption in GDTM, `ADMN` localizes within $\sim 2$ cm of the Upper Bound while reducing latency by $\sim 60\%$ and FLOPs by $\sim 75\%$.

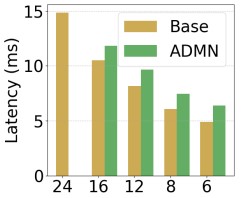 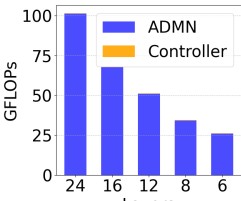 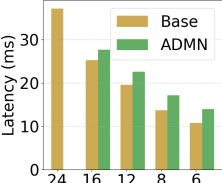 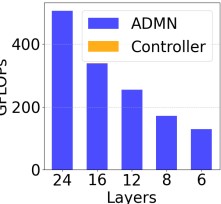

Figure 4: Latency (ms) and GFLOPs vs Layers for GDTM (left) and MM-Fi (Right)

## 4.3 Extended Evaluations

**Three Modalities.** To ensure that `ADMN` generalizes to more than two modalities, we perform localization on GDTM with RGB, depth, and mmWave radar modalities. Figure 7 shows a CDF of the error on a dataset where each modality has a 50% likelihood of suffering extreme Gaussian Noise. `ADMN` continues to perform correct allocations with three heterogeneous modalities.

**Unequal Backbone Computation.** For each AVE dataset sample, we process eight image frames and one large audio spectrogram. Thus, the visual backbone consumes approximately *three times* the FLOPs of the audio backbone. We neglected this in Table 3 by assuming that all backbones were equally resource intensive. In Table 4, we present results in terms of *Audio Layer Budgets*, where activating one layer in the image backbone costs *three audio layers*. In comparison to Naive Alloc., which activates the same number of **real** layers per modality (e.g., Naive Alloc. 12 layers is three image and three audio layers), `ADMN` performs superior allocations accounting for unequal computation.

Table 4: Unequal AVE Classification Accuracy (↑), budget in Audio Layers

| Method | AVE (Rain + Background Noise) | | | |
| | 12 | 16 | 24 | 32 |
| --- | --- | --- | --- | --- |
| Upper Bound | 71.19% | | | |
| `ADMN` | 51.83% | 64.34% | 66.46% | 67.68% |
| Naive Alloc. | 36.16% | 46.89% | 65.71% | 67.95% |

Table 5: Impact of supervision on AVE Classification Accuracy (↑)

| Method | AVE (Rain + Background Noise) | | | |
| | 6 | 8 | 12 | 16 |
| --- | --- | --- | --- | --- |
| Upper Bound | 71.19% | | | |
| `ADMN` | 57.07% | 62.95% | 67.48% | 66.60% |
| `ADMN_AE` | 52.95% | 62.30% | 66.77% | 66.67% |
| Task Loss `ADMN` | 46.20% | 57.13% | 64.53% | 67.50% |

## 4.4 Ablation Studies

**Efficacy of LayerDrop.** LayerDrop is integrated into two stages – initial MAE pretraining on ImageNet and Stage 1 Finetuning. Figure 6 showcases the localization error on the GDTM dataset for separate unimodal depth and image networks. We observe that adding Layer-Drop during finetuning (LD FT) has the greatest impact, but benefit is observed when added into both stages (LD Both). The ability to drop many layers without significant degradation confirms that LayerDrop is compatible with the ViT architecture, MAE pretraining, and is also effective when finetuned on another task. We present additional results in Appendix A.4.

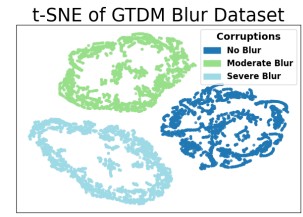

Figure 5: t-SNE of the autoencoder for different levels of RGB Blur on GDTM Blur

**Impact of Controller Supervision.** In Table 5, we omit both the autoencoder pretraining and the corruption supervision to get a variant of `ADMN` supervised purely by the Task Loss only. The inferior results show the importance of utilizing either corruption supervision or autoencoder pretraining.

**Autoencoder Latent Space.** In Figure 5, we visualize how `ADMN_AE`'s autoencoder performs corruption-aware clustering in the latent space through a t-SNE plot. We note that this behavior is learned purely through a data driven manner, and *no information on the corruptions were provided during autoencoder training*. The obvious clustering of the various corruptions reveals how `ADMN_AE` is an effective replacement for explicit corruption supervision.

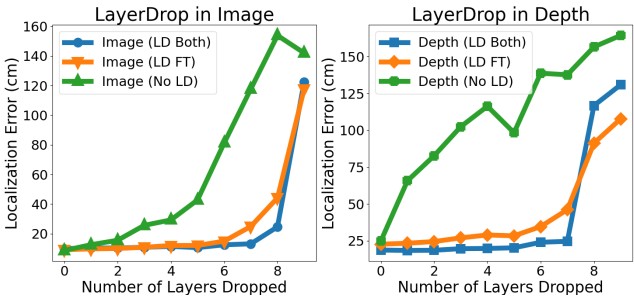
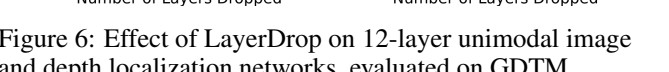
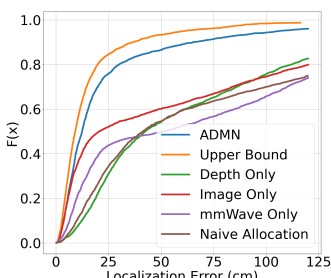

Figure 6: Effect of LayerDrop on 12-layer unimodal image and depth localization networks, evaluated on GDTM.

Figure 7: Localization Error for GDTM with Three Modalities

## 5 Limitations and Discussions

**Dynamic Compute Constraints:** One drawback is that the `ADMN` controller is trained only for one particular layer budget $L$. Consequently, we must train and load multiple controllers for *dynamic test-time compute requirements* induced by factors such as thermal throttling. Nevertheless, the training overhead for the controllers is low (Section A.1), enabling a controller to be trained for every layer budget. The controller also constitutes only 2.3% and 3.2% of the total network parameters in the GDTM and MM-Fi tasks, respectively, allowing for easy storage on disk. Furthermore, we present some preliminary results on a universal controller in Appendix A.1.

**Batched Inference:** Since `ADMN` obtains the layer allocation prior to backbone execution, future work can exploit this for efficient batched inference. Classic adaptive models are incompatible with batched inference due to per-layer execution decisions for each sample. However, with `ADMN`, we can run an initial profiling stage and group samples into *sub-batches* based on similar layer allocation. For instance, samples with high depth noise and low image noise activate similar layers and can be grouped together.

**Fusion with Early Exit:** While `ADMN` and unimodal Early-Exit methods tackle fundamentally different problems, the two techniques can be combined for further efficiency gains. `ADMN` always allocates $L$ layers across all the modalities. However, on simple inputs, all $L$ layers may not be necessary, allowing for Early-Exit techniques to improve performance.

## 6 Conclusion

This paper proposes `ADMN`, a multimodal network capable of dynamically adjusting the number of active Transformer layers across modalities according to the quality of each sample's input modalities. Through this continuous reallocation, `ADMN` can match the accuracy of far larger networks while utilizing a fraction of their operations. Additionally, the dynamic backbones of `ADMN` are also well suited for scenarios with adaptive compute, ranging from heterogeneous deployment devices to fluctuating energy availability. We demonstrate the superiority of `ADMN` compared to other baselines across both classification and localization tasks.

## 7 Acknowledgments

The research reported in this paper was sponsored in part by the DEVCOM Army Research Laboratory (award # W911NF1720196 ), the Air Force Office of Scientific Research (awards # FA95502210193 and FA95502310559), and the National Institutes of Health (award # 1P41EB028242). The views and conclusions contained in this document are those of the authors and should not be interpreted as representing the official policies, either expressed or implied, of the funding agencies. Jason Wu was supported by the Department of Defense (DoD) through the National Defense Science & Engineering Graduate (NDSEG) Fellowship Program. Mani Srivastava was also partially supported by the Mukund Padmanabhan Term Chair at UCLA.

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

# A   Appendix

## A.1   Controller Details

**Training Details and Overhead.** We found it sufficient to train the controller for 10 and 15 epochs on the localization and classification tasks, respectively. We use a learning rate of $1 \times 10^{-3}$ and a linear decay scheduler. For the corruption supervised `ADMN` controller, we train only on the corruption loss for the first epoch to ensure that the model could accurately differentiate between modality QoI before learning optimal layer allocations. After the first epoch, we add the task loss and optimize the controller over the joint loss. Training an `ADMN` controller on the GDTM Gaussian Noise dataset took only 27 minutes on an Nvidia RTX 4090. For `ADMN_AE`, we divide the training process into two stages. First, we train the autoencoder through reconstruction loss on the target dataset. This is a one-time process, and the set of trained encoder weights can then be used to train controllers of all layer budgets. Subsequently, we load the encoder weights into the controller, freeze them to maintain the QoI information, and train only the layer MLP of the controller on the task loss.

| Controller Type | Seed | 6 Layers | 8 Layers | 12 Layers | 16 Layers |
|---|---|---|---|---|---|
| `ADMN` | 100 | 51.96 | 36.35 | 32.24 | 29.38 |
| | 200 | 47.29 | 39.12 | 33.01 | 30.95 |
| | 300 | 56.09 | 37.02 | 32.11 | 31.27 |
| **Straight-Through** | 100 | 118.14 | 90.36 | 73.87 | 32.15 |
| | 200 | 94.43 | 58.66 | 50.76 | 51.32 |
| | 300 | 103.21 | 82.20 | 51.84 | 29.81 |

Table 6: `ADMN` GDTM Gaussian Noise Localization Error (cm) $\downarrow$ comparing `ADMN` and the Straight-Through Estimator, with three seeds each

**Universal Controller.** On the GDTM dataset with Gaussian Noise corruption, we train a single controller for all four layer allocations (6, 8, 12, and 16 layers). We introduce 4 learnable context tokens that correspond to each layer budget, allowing the controller to understand the current layer budget. During training, we randomly select a layer budget for each batch, and concatenate the relevant context token to the input of the TE Fusion block within the controller.

Table 7 depicts the performance of the universal controller (single set of weights) in comparison to training a new controller for each layer allocation on the Gaussian Noise corrupted GDTM dataset. The universal controller successfully accommodates different layer budgets, even outperforming the baseline `ADMN` on small layer allocations, at the cost of slightly worse performance on larger layer allocations. Future work can further explore the design of the controller for elevated performance.

**End-to-end Controller Training.** We provide insights into why corruption-aware supervision or autoencoder initialization is necessary during controller training. We hypothesize that the *complexity of the training process* hinders the controller from learning the corruption distribution without any corruption supervision (explicit with metadata or through autoencoder). Since the selection of a layer is not a differentiable operation, we model it with Gumbel-Softmax Sampling, followed by discretization and a straight through estimator. The gradients received by the controller are thus only an estimation of how that particular layer impacts the downstream loss, with additional complexity arising from the dependence on other layers that were selected alongside it. Consequently, despite training the layer selection mechanism end-to-end, it is very difficult for the earlier perceptual components of the controller to learn to attend to the input modality QoI from the noisy layer gradient information, usually requiring assistance in the form of our corruption-aware supervision or autoencoder initialization.

## A.2   Justification of Gradient Propagation Technique in the Controller

**Directly Employing the Straight-Through Estimator.** `ADMN` utilizes the combination of standard temperature Gumbel-Softmax sampling and the straight-through estimator to propagate gradients over the discretization to the continuous logits. One natural question is whether Gumbel-Softmax Sampling is necessary, as one could theoretically simply discretize the raw logits and propagate gradients with the straight-through estimator. In Table 6, we present the localization results on the GDTM Gaussian Noise dataset across different layer configurations, with three seeds for each

| Method | 6 Layers | 8 Layers | 12 Layers | 16 Layers |
|---|---|---|---|---|
| ADMN | 51.4 | 39.0 | 33.1 | 30.3 |
| ADMN **Universal** | 43.77 | 36.78 | 36.69 | 34.94 |

Table 7: Localization error on the GDTM Gaussian Noise. Comparison between base `ADMN` and `ADMN` with a single universal controller

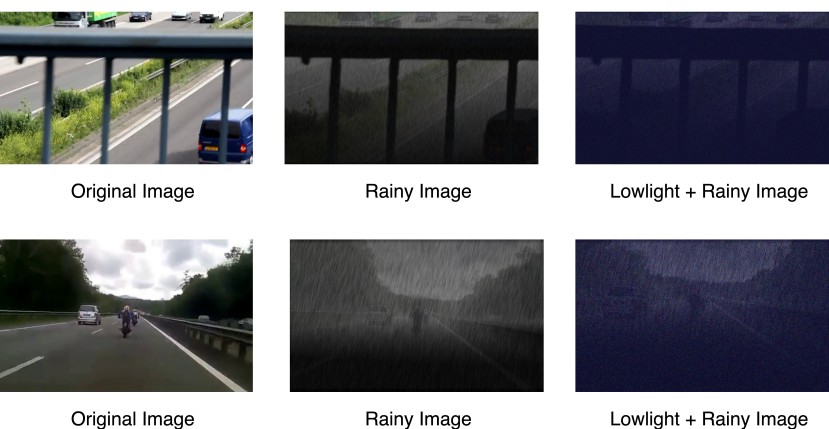

Figure 8: Visual Depiction of AVE image corruption for outdoor classes

experiment. The results highlight that Gumbel-Softmax sampling plays an important role in model training.

This behavior can be attributed to several reasons. First, by applying the softmax function to the logits, we convert them into probability values where one logit's high probabilities come at the expense of the others. As a result, the softmax function encourages the controller to select only the $L$ best performing layers for some value of noise and minimize the probability of the remaining layers. Additionally, utilizing the Gumbel distribution also introduces *stochasticity* into the sampling process. Instead of always selecting the top-L logits as the active layers, the stochasticity intuitively serves to encourage *exploration* of different layer configurations.

**Progressive Top-L Gumbel Softmax Sampling.** Instead of employing the straight-through estimator, one can also utilize repeated Gumbel-Softmax Sampling to emulate discrete top-L sampling. Xie et. al. [33] proposed a method to emulate discrete top-L sampling by repeatedly applying the softmax function $L$ times while adjusting the logits each iteration. However, these methods may cause issues when applied to `ADMN`. First, methods utilizing Gumbel-Softmax Sampling to emulate discrete distributions typically have to undergo *temperature annealing* [15], where the temperature is slowly decreased until the distribution is approximately categorical. Utilizing annealing can lead to a longer and more complicated training process for the controller. Additionally, the lack of explicit discretization during training may also result in a distribution shift at inference time, where the controller may learn to over-rely upon partially activated layers during training.

### A.3  Additional Dataset Information

We provide further information on the construction of the datasets used in our main evaluations. Visual examples of samples from these datasets, along with the corresponding controller layer allocations, are shown in Section A.7.

**GDTM Gaussian Noise.** We define four standard deviations for each modality. The image modality defines a set of standard deviations $[0, 1, 2, 3]$ while the depth modality defines its own set of standard deviations $[0, 0.25, 0.5, 0.75]$. We construct the dataset by randomly selecting a standard deviation from each set to corrupt the corresponding modality with. Note that since the GDTM dataset is *distributed* with three nodes, we apply the same corruptions across the three nodes. Thus, the same

noise that is applied to RGB image in Node 1 will be applied to RGB image in Nodes 2 and 3. We do not consider QoI variations across the distributed nodes.

**GDTM Lowlight.** We define three levels of lowlight - ordinary lighting, medium lowlight, and severe lowlight. Under ordinary lowlight, we do not modify the RGB modality, but introduce saturation corruption into the depth modality. IR depth cameras frequently saturate under direct lighting conditions, and we wished to emulate this phenomenon in the ordinary lighting case. We obtained a sample of corrupted depth data (sensor is entirely saturated and contains no useful information) by bringing the sensor into direct sunlight. Under medium lowlight, the RGB image is darkened while the depth frame is left unchanged. Finally, for severe lowlight, the RGB image is substantially darkened, and the depth frame is left unchanged. The dataset is equally split between these three conditions. Similarly to the GDTM Gaussian Noise dataset, there are no QoI variations across distributed nodes - a given sensor type across all nodes will suffer from the same lighting condition.

**GDTM Blur.** We simulate the effects of blur on only the RGB camera modality. We define three levels of blur - no blur, medium blur, and severe blur through the use of Gaussian Kernel blurring. We split the dataset equally among these three combinations with depth left unchanged. The blur corruption does not change across the distributed nodes.

**MM-Fi Gaussian Noise.** MM-Fi Gaussian Noise leverages a similar principle to the GDTM Gaussian Noise dataset. The image modality defines a set of standard deviations $[0, 0.75, 1.5, 2]$ while the depth modality defines a set of standard deviations $[0, 2, 3, 4]$. The MM-Fi dataset contains data across the *temporal axis* with several frames spanning a given human activity. When we draw one standard deviation for a modality, we apply noise with that one standard deviation across *all frames of that given modality*. Within a particular sample, we do not consider QoI variations across frames.

**AVE Rain/Background Noise.** The audiovisual AVE dataset contains 28 classes that contain both indoor and outdoor samples. To ensure that we apply realistic corruptions, we corrupt indoor and outoor samples differently. For outdoor samples, we choose to induce rain and lowlight corruption in the RGB image modality, and add background wind noise to corrupt the audio modality. For indoor samples, the RGB modality is corrupted with lowlight, and audio is corrupted with background noise of a standing fan. We follow the techniques in [31] for adding rain, and mix in the background audio noise through the PyDub library. Each clean outdoor/indoor sample is transformed into one with corrupted image/clean audio, and another with clean image/corrupted audio. We show the rain+lowlight corruption of outdoor samples in Figure 8 to demonstrate why we had to add an additional lowlight corruption. After adding rain, the main subject of the image is still quite visible, necessitating the addition of lowlight to ensure that the model cannot rely solely on visual information to perform classification.

## A.4 Additional LayerDrop Results

| Removed Layer Indices | Normal Pre + Normal FT | LayerDrop Pre + Normal FT | Normal Pre + LayerDrop FT | LayerDrop Pre + LayerDrop FT |
|---|---|---|---|---|
| None | 81.16% | 80.36% | 79.91% | 78.92% |
| 6 | 79.99% | 79.35% | 79.61% | 78.59% |
| 6, 8 | 77.20% | 76.77% | 78.68% | 77.72% |
| 4, 6, 8 | 72.97% | 73.94% | 77.64% | 76.84% |
| 2, 4, 6, 8 | 70.01% | 71.48% | 76.90% | 76.37% |
| 2, 4, 6, 8, 10 | 64.27% | 65.07% | 75.44% | 74.70% |
| 2, 4, 6, 7, 8, 10 | 33.91% | 38.76% | 69.83% | 69.28% |
| 1, 2, 4, 6, 7, 8, 10 | 13.74% | 23.60% | 66.67% | 66.94% |

Table 8: ImageNet-1K performance with different layer indices removed. Normal refers to a LayerDrop rate of 0, while LayerDrop refers to utilizing a LayerDrop rate of 0.2. Pre indicates the MAE pretraining stage, while FT refers to supervised finetuning on ImageNet-1K.

**ImageNet-1K Results.** The visual backbones are first pretrained on the ImageNet dataset with Masked Autoencoder pretraining, in which we add LayerDrop. To understand its performance on the ImageNet-1K dataset, we perform a subsequent stage of supervised learning on the ImageNet dataset to obtain the validation accuracy. Table 8 reveals the validation accuracy on the ImageNet-1K dataset with various dropped layers, and with LayerDrop integrated into different stages of training. When comparing the model trained without any usage of LayerDrop to the one in which

| Removed Layer Indices | Normal Pre + Normal FT | LayerDrop Pre + LayerDrop FT |
|---|---|---|
| None | 33.77% | 32.01% |
| 6 | 31.31% | 31.67% |
| 6, 8 | 27.43% | 31.17% |
| 4, 6, 8 | 22.81% | 30.60% |
| 2, 4, 6, 8 | 15.64% | 30.08% |
| 2, 4, 6, 8, 9 | 9.82% | 28.79% |
| 2, 4, 6, 7, 8, 9 | 4.42% | 24.64% |
| 1, 2, 4, 6, 7, 8, 9 | 2.43% | 22.77% |
| 1, 2, 4, 5, 6, 7, 8, 9 | 1.57% | 14.41% |
| 1, 2, 3, 4, 5, 6, 7, 8, 9 | 1.11% | 3.89% |
| 1, 2, 3, 4, 5, 6, 7, 8, 9, 10 | 1.04% | 2.20% |

Table 9: AudioSet performance with different layer indices removed. Normal refers to a LayerDrop rate of 0, while LayerDrop refers to utilizing a LayerDrop rate of 0.2. Pre indicates the MAE pretraining stage, while FT refers to supervised finetuning on AudioSet.

LayerDrop was employed in both stages, we can observe an accuracy improvement of over 50% when 7 layers are dropped during inference time. Curiously, given that LayerDrop is added during supervised finetuning, applying MAE pretraining with LayerDrop does not appear to be necessary in ImageNet-1K. However, the results in Figure 6 showcase that it has an impact on downstream tasks.

**AudioSet-2M Results.** The audio backbones are pretrained on the AudioSet dataset with AudioMAE pretraining. We present the AudioSet test split results in Table 9 after finetuning on a balanced subset referred to as AudioSet-20k. We compare the impact of adding LayerDrop into both stages of the process, and find that introducing LayerDrop is key for graceful degradation as layers are removed. With four layers removed, the model pretrained and finetuned without LayerDrop has its accuracy cut in half, while the model with LayerDrop loses less than 10% accuracy.

| Noise Type | Layer | Random Seed | | | | | | Avg. | Std. |
|---|---|---|---|---|---|---|---|---|---|
| | | 100 | 200 | 300 | 400 | 500 | 600 | | |
| Gaussian | 6 | 52.0 \| 53.3 | 47.3 \| 53.7 | 56.1 \| 54.0 | 52.3 \| 53.6 | 55.2 \| 53.5 | 45.6 \| 53.4 | 51.4 \| 53.6 | 4.2 \| 0.2 |
| | 8 | 36.4 \| 37.8 | 39.1 \| 37.6 | 37.0 \| 37.5 | 41.4 \| 37.9 | 41.0 \| 41.7 | 38.8 \| 38.1 | 39.0 \| 38.4 | 2.0 \| 1.6 |
| | 12 | 32.2 \| 35.0 | 33.0 \| 32.6 | 32.1 \| 35.0 | 32.8 \| 32.6 | 35.8 \| 32.1 | 32.5 \| 33.5 | 33.1 \| 33.5 | 1.4 \| 1.3 |
| | 16 | 29.4 \| 29.7 | 31.0 \| 30.2 | 31.3 \| 30.6 | 29.3 \| 28.7 | 31.6 \| 28.4 | 29.3 \| 28.6 | 30.3 \| 29.4 | 1.1 \| 0.9 |
| Lowlight | 6 | 56.8 \| 33.7 | 34.0 \| 40.5 | 36.6 \| 33.4 | 54.0 \| 36.8 | 57.8 \| 33.5 | 57.5 \| 33.5 | 49.5 \| 35.2 | 11.1 \| 2.9 |
| | 8 | 19.5 \| 19.4 | 27.2 \| 27.0 | 27.2 \| 19.8 | 22.4 \| 19.0 | 27.5 \| 23.4 | 19.5 \| 27.9 | 23.9 \| 22.7 | 3.9 \| 4.0 |
| | 12 | 18.1 \| 18.2 | 17.6 \| 18.2 | 18.1 \| 18.2 | 18.2 \| 20.3 | 18.4 \| 18.3 | 17.6 \| 18.7 | 18.0 \| 18.7 | 0.3 \| 0.8 |
| | 16 | 17.3 \| 18.1 | 17.5 \| 17.6 | 16.9 \| 17.3 | 17.1 \| 17.4 | 17.4 \| 17.7 | 17.4 \| 17.7 | 17.3 \| 17.6 | 0.2 \| 0.3 |
| Blur | 6 | 14.7 \| 12.6 | 11.2 \| 25.2 | 11.2 \| 12.2 | 11.2 \| 11.3 | 11.5 \| 11.4 | 11.2 \| 11.3 | 11.8 \| 14.0 | 1.4 \| 5.5 |
| | 8 | 11.0 \| 10.6 | 10.6 \| 11.1 | 11.1 \| 11.3 | 12.9 \| 10.5 | 11.1 \| 13.1 | 10.5 \| 11.7 | 11.2 \| 11.4 | 0.9 \| 1.0 |
| | 12 | 10.0 \| 9.6 | 9.2 \| 10.2 | 9.3 \| 9.8 | 9.4 \| 9.6 | 10.5 \| 9.7 | 10.0 \| 9.7 | 9.7 \| 9.7 | 0.5 \| 0.2 |
| | 16 | 10.4 \| 9.2 | 9.6 \| 9.8 | 9.6 \| 9.6 | 10.1 \| 9.6 | 9.6 \| 9.8 | 10.0 \| 9.6 | 9.9 \| 9.6 | 0.3 \| 0.2 |

Table 10: `ADMN` (left) vs `ADMN_AE` (right) Localization Error (cm) on GDTM Dataset

| Layer | Random Seed | | | | | | Avg. | Std. |
|---|---|---|---|---|---|---|---|---|
| | 100 | 200 | 300 | 400 | 500 | 600 | | |
| 6 | 34.3 \| 38.0 | 36.1 \| 37.7 | 34.6 \| 36.7 | 35.2 \| 35.2 | 33.6 \| 33.6 | 36.4 \| 35.8 | 35.0 \| 36.2 | 1.1 \| 1.6 |
| 8 | 38.9 \| 37.7 | 40.7 \| 35.8 | 42.0 \| 37.3 | 39.5 \| 39.5 | 39.2 \| 39.2 | 35.2 \| 41.1 | 39.2 \| 38.4 | 2.3 \| 1.9 |
| 12 | 39.8 \| 40.1 | 42.6 \| 43.5 | 42.0 \| 44.1 | 46.6 \| 44.8 | 39.8 \| 40.4 | 40.7 \| 40.1 | 41.9 \| 42.2 | 2.6 \| 2.2 |
| 16 | 42.6 \| 43.8 | 42.6 \| 46.3 | 42.0 \| 42.9 | 46.0 \| 47.8 | 43.5 \| 42.3 | 43.2 \| 43.2 | 43.3 \| 44.4 | 1.4 \| 2.2 |

Table 11: `ADMN` (left) vs `ADMN_AE` (right) Classification Accuracy (%) on MM-Fi Dataset

## A.5 Stability Analysis Under Noise and Seeds

To assess the stability of our results across random initializations, we train the `ADMN` and `ADMN_AE` controllers across six random seeds for each layer budgets and dataset. Tables 10, 11, 12, showcase the result for each seed, along with the averaged result and standard deviation:

| | | | Random Seed | | | | Avg. | Std. |
|---|---|---|---|---|---|---|---|---|
| Layer | 100 | 200 | 300 | 400 | 500 | 600 | | |
| 6 | 59.4 \| 49.6 | 51.9 \| 56.4 | 56.9 \| 59.2 | 58.2 \| 44.5 | 57.6 \| 55.1 | 58.5 \| 52.9 | 57.1 \| 53.0 | 2.7 \| 5.3 |
| 8 | 61.5 \| 65.2 | 57.0 \| 62.7 | 66.3 \| 61.7 | 65.6 \| 59.5 | 61.8 \| 65.0 | 65.5 \| 59.7 | 62.9 \| 62.3 | 3.6 \| 2.5 |
| 12 | 67.0 \| 66.0 | 66.1 \| 64.3 | 61.6 \| 67.3 | 69.0 \| 70.3 | 70.7 \| 65.6 | 70.6 \| 67.1 | 67.5 \| 66.8 | 3.4 \| 2.0 |
| 16 | 67.1 \| 66.3 | 69.0 \| 68.2 | 67.3 \| 68.1 | 61.5 \| 66.3 | 66.8 \| 63.6 | 68.0 \| 67.5 | 66.6 \| 66.7 | 2.6 \| 1.7 |

Table 12: `ADMN` (left) vs `ADMN_AE` (right) Classification Accuracy (%) on AVE Dataset

**ADMN vs ADMN_AE:** When comparing `ADMN` and `ADMN_AE`, one may expect that `ADMN` should provide more consistent results, given that we explicitly leverage known ground truth labels as QoI supervision. However, in the case of 6 seeds GDTM Lowlight, we find that the `ADMN_AE` approach provides a significantly lower standard deviation with better localization results. This can potentially be attributed towards the freezing of network weights employed by `ADMN_AE`, where we only learn the layer MLP during controller training. In contrast, `ADMN`'s controller is fully unfrozen during the training process, resulting in greater susceptibility to poor seed initializations. In the rest of the experiments, the two methods have approximately similar standard deviations.

**Layer Budget:** We observe a consistent reduction of standard deviation with increasing layer budget, which suggests that larger layers budgets have more stable layer allocation policies. This aligns with the intuition that the result is more sensitive to layer allocation strategies when there is a smaller budget.

## A.6 Training Details

| Parameter | MM-Fi | GDTM | Lowlight | Blur | AVE |
|---|---|---|---|---|---|
| Epochs | 400 | 400 | 100 | 200 | 200 |
| Learning Rate | 1E-4 | 5.00E-04 | 5.00E-04 | 5.00E-04 | 5.00E-04 |
| Scheduler | LinearLR | — | — | — | — |
| Optimizer | Adam | Adam | Adam | Adam | Adam |
| LayerDrop | 0.2 | 0.2 | 0.2 | 0.2 | 0.4 |
| Fusion Layers | 6 | 6 | 6 | 6 | 6 |
| Fusion Dimension | 64 | 256 | 256 | 256 | 256 |
| Fusion Heads | 4 | 4 | 4 | 4 | 4 |
| Modality Dropout | 0.1 | 0.1 | 0.1 | 0.1 | 0.1 |
| Depth Corruption | Gaussian Noise | Gaussian Noise | Lowlight | Blur | — |
| Image Corruption | Gaussian Noise | Gaussian Noise | Saturation | — | Rain/Lowlight |
| Audio Corruption | — | — | — | — | Wind/Fan |

Table 13: Finetuning Settings of All Datasets

**Finetuning Details.** We depict the finetuning configurations for each dataset in Table 13. We employ the embedding-level fusion architecture in all these datasets, where we extract unimodal features with modality specific backbones, merge them with a transformer encoder, and perform the task with an output head. The training configurations are similar across the datasets, with small exceptions in MM-Fi and AVE. In MM-Fi, we found it advantageous to utilize a fusion dimension of 64 instead of 256 due to the large number of image and depth frames that we process. In the AVE dataset, we employed a more aggressive LayerDrop rate of 0.4 during the fine-tuning process, as we found that the standard rate of 0.2 led to significant and rapid degradation when dropping layers during inference time.

**Compute Requirements.** Pretraining on ImageNet-1k and AudioSet-2M with the MAE and AudioMAE techniques were performed on a GPU server containing 4 Nvidia H100 GPUs over 5 days. Finetuning the models and performing controller training were performed on two machines - one with an Nvidia 4090 and the other with two Nvidia 3090 GPUs.

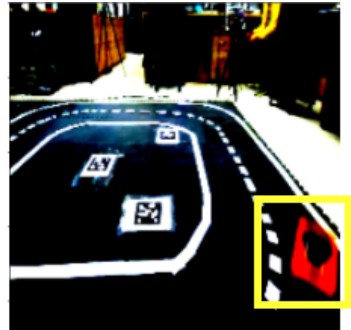

Image Noise Standard
Deviation: **0**

Selected Layers:
**[1** 0 0 0 **1** 0 **1** 0 **1** 0 0 **1]**

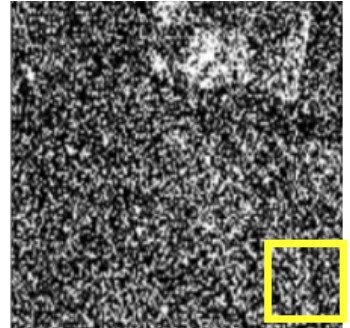

Depth Noise Standard
Deviation: **0.75**

Selected Layers:
**[1** 0 0 0 0 0 0 0 0 0 0 0 **0]**

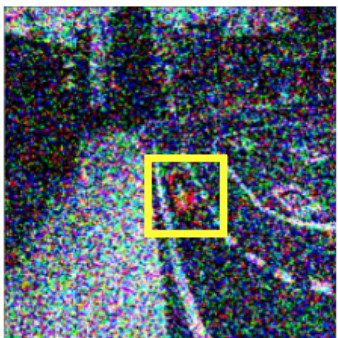

Image Noise Standard
Deviation: **1**

Selected Layers:
**[1** 0 0 0 **1 1 1** 0 0 0 0 **1]**

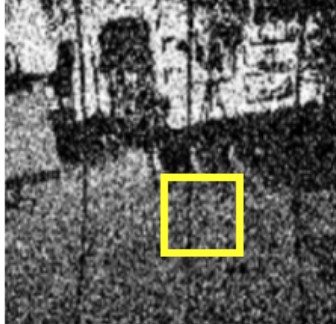

Depth Noise Standard
Deviation: **0.25**

Selected Layers:
**[1** 0 0 0 0 0 0 0 0 0 0 0 **0]**

## A.7    Qualitative Results

In Figures 9, 10, 11, 12, 14, we visually showcase examples of the corrupted data samples and the appropriate controller allocations for the GDTM Gaussian Noise, GDTM Lowlight, GDTM Blur, AVE Corruption datasets, and MM-Fi Gaussian Noise datasets, respectively. With the exception of the AVE Corruption dataset, we show the normalized images that are input into the model. The controller always activates the first backbone layer of each modality for stability reasons. We observe that the controller makes intelligent allocation decisions that align with our expectations from the modality QoI. One interesting phenomenon occurs in the GDTM blur dataset, where although the vehicle is not visible in the "Heavy Blur" image, the controller still allocates all the resources to image, as the model has superior perception compared to human vision and can localize the car in spite of the heavy blur.

Additionally, we also compare the ADMN controller to the Task Loss controller (Figure 13) that is supervised by neither the corruption nor the autoencoder pretraining. On the AVE dataset, we can

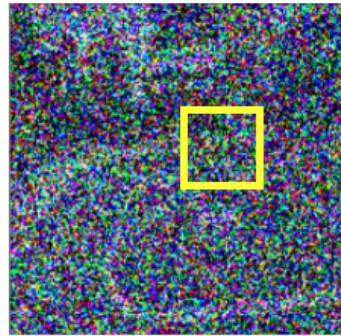

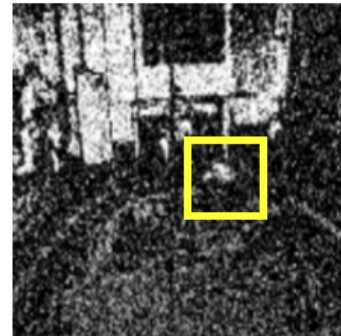

Image Noise Standard
Deviation: **3**

Depth Noise Standard
Deviation: **0.25**

Selected Layers:
**[1** 0 0 0 0 **1** 0 0 0 0 0 **1]**

Selected Layers:
**[1** 0 0 0 0 0 **1 1** 0 0 0 0**]**

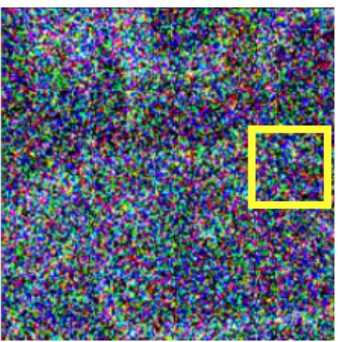

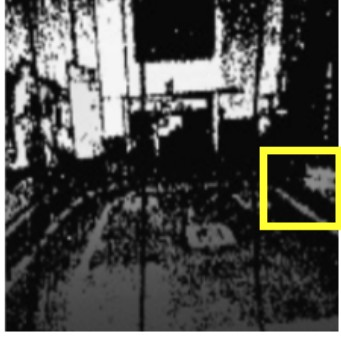

Image Noise Standard
Deviation: **3**

Depth Noise Standard
Deviation: **0**

Selected Layers:
**[1** 0 0 0 0 0 0 0 0 0 0 0**]**

Selected Layers:
**[1** 0 0 **1** 0 0 0 **1** 0 **1 1]**

Figure 9: Visual Results on the GDTM Dataset highlighting the impact of noise and featuring the controller layer allocation

see that the controller allocations do not account for the changing modality QoI, which explains the inferior results in Table 5.

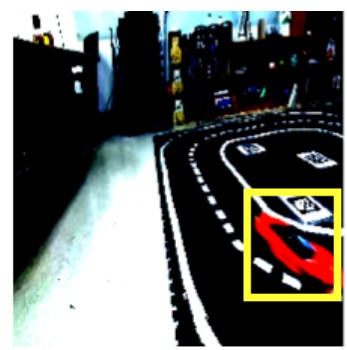

Clean Image

Selected Layers:
[**1** 0 0 0 0 **1 1 1** 0 0 0 **1**]

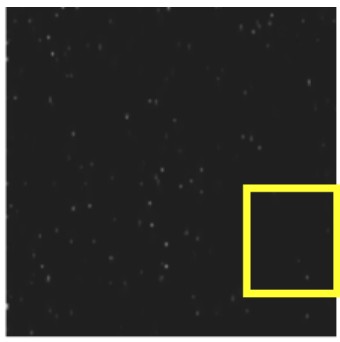

Corrupted Depth

Selected Layers:
[**1** 0 0 0 **1** 0 0 0 0 0 **1** 0]

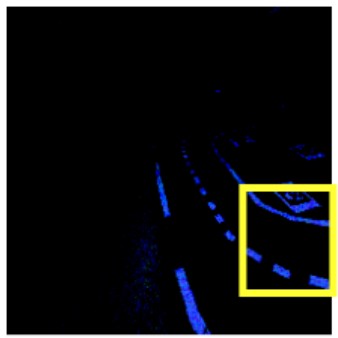

Medium Lowlight

Selected Layers:
[**1** 0 0 0 0 0 0 0 0 0 0 **1**]

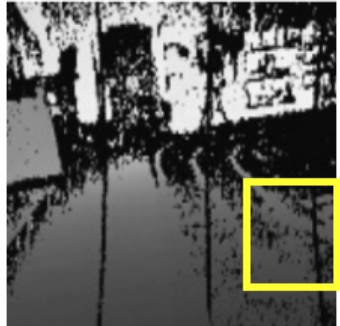

Clean Depth

Selected Layers:
[**1** 0 0 0 **1 1 1** 0 0 0 **1 1**]

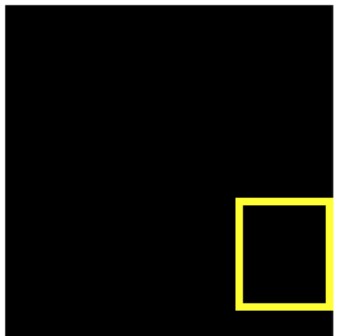
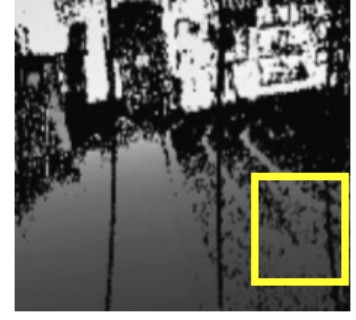

Severe Lowlight | Clean Depth

Selected Layers: | Selected Layers:
[1 0 0 0 0 0 0 0 0 0 0 1] | [1 0 0 0 1 1 1 0 0 0 1 1]

Figure 10: Visual Results on the GDTM Dataset highlighting the impact of lowlight and featuring the controller layer allocation

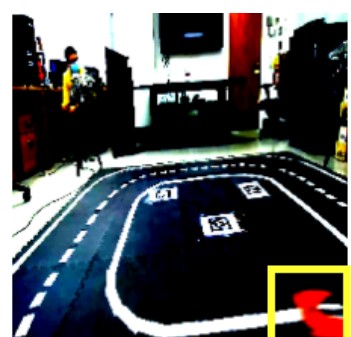

Clean Image | Clean Depth

Selected Layers: | Selected Layers:
[1 0 0 1 1 1 1 0 0 0 1] | [1 0 0 0 0 0 0 0 0 0 0 0]

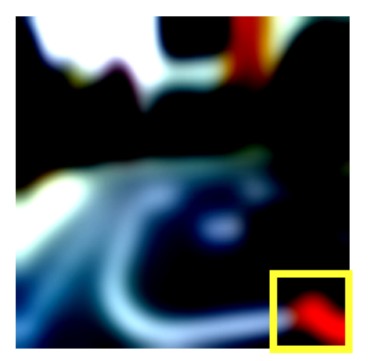
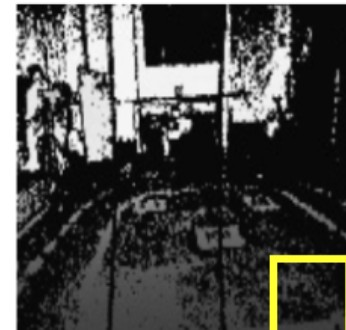

Medium Blur Image

Selected Layers:
[1 0 0 1 1 1 1 1 0 0 0 1]

Clean Depth

Selected Layers:
[1 0 0 0 0 0 0 0 0 0 0 0]

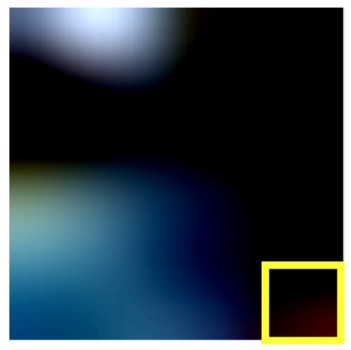
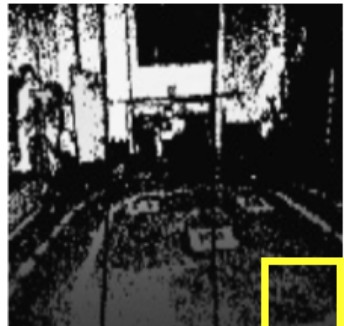

Heavy Blur Image

Selected Layers:
[1 0 0 1 1 1 1 1 0 0 0 1]

Clean Depth

Selected Layers:
[1 0 0 0 0 0 0 0 0 0 0 0]

Figure 11: Visual Results on the GDTM Dataset highlighting the impact of blur and featuring the controller layer allocation

Corrupted Audio &
Clean Image

Selected Layers:
[**1** 0 0 0 0 0 **1** 0 0 0 0 **1**]

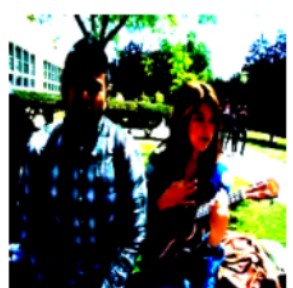

Selected Layers:
[**1** 0 0 0 0 **1 1 1** 0 0 0 **1**]

Clean Audio &
Corrupted Image

Selected Layers:
[**1** 0 **1** 0 **1** 0 **1 1** 0 0 0 **1**]

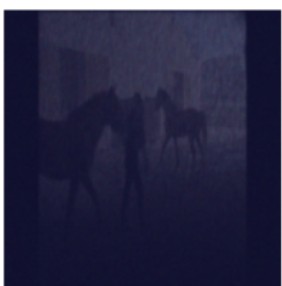

Selected Layers:
[**1** 0 0 0 0 0 0 0 0 0 0 **1**]

Clean Audio &
Corrupted Image

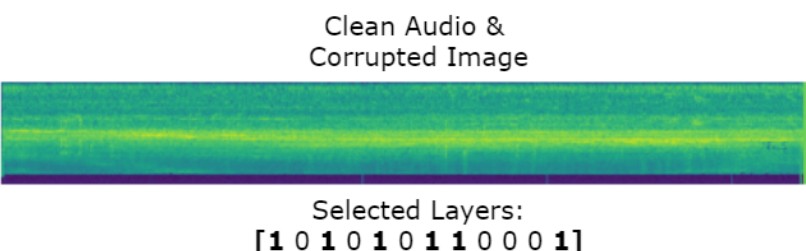

Selected Layers:
[**1** 0 **1** 0 **1** 0 **1 1** 0 0 0 **1**]

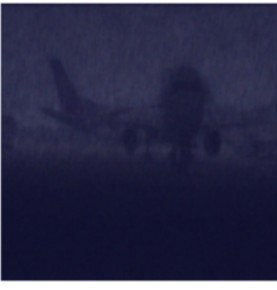

Selected Layers:
[**1** 0 0 0 0 0 0 0 0 0 0 0 **1**]

Figure 12: Visual Results on the AVE Dataset highlighting the rain/lowlight visual corruptions and background noise audio corruptions. We also showcase the ADMN controller layer allocation.

Corrupted Audio &
Clean Image

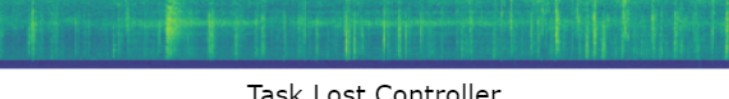

Task Lost Controller
Selected Layers:
[**1** 0 0 **1** 0 0 0 0 0 0 0 **1**]

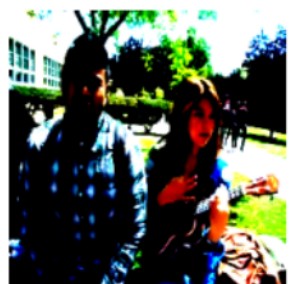

Task Loss Controller
Selected Layers:
[**1** 0 0 0 0 **1** 0 **1** 0 0 **1** **1**]

Clean Audio &
Corrupted Image

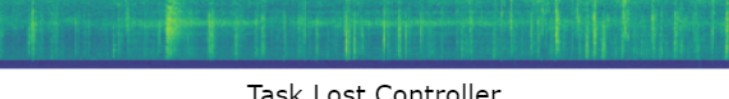

Task Lost Controller
Selected Layers:
[**1** 0 0 0 0 0 0 0 **1** 0 0 **1**]

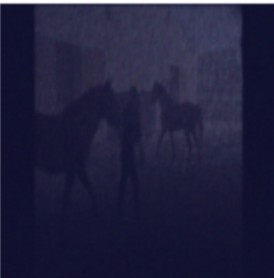

Task Lost Controller
Selected Layers:
[**1** 0 0 0 0 **1** 0 **1** 0 0 **1** **1**]

Figure 13: Visual Results on the AVE Dataset highlighting the rain/lowlight visual corruptions and
background noise audio corruptions. We showcase the **Task Loss** controller layer allocation.

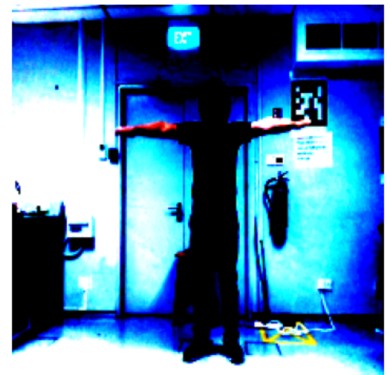

Image Noise Standard
Deviation: 0.0

Selected Layers:
[1 0 0 0 1 0 1 1 1 0 0 1]

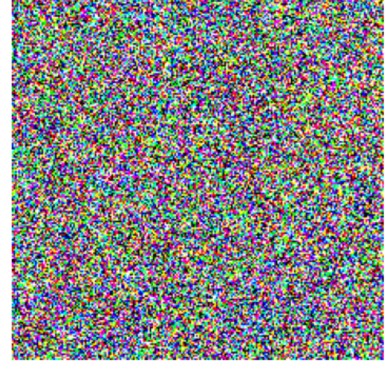

Depth Noise Standard
Deviation: 4.0

Selected Layers:
[1 0 0 0 1 0 0 0 0 0 0 0]

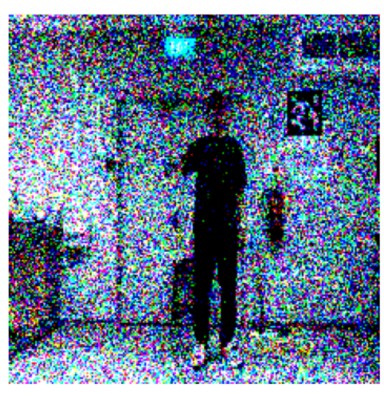

Image Noise Standard
Deviation: 0.75

Selected Layers:
[1 0 0 0 1 0 1 1 1 0 0 1]

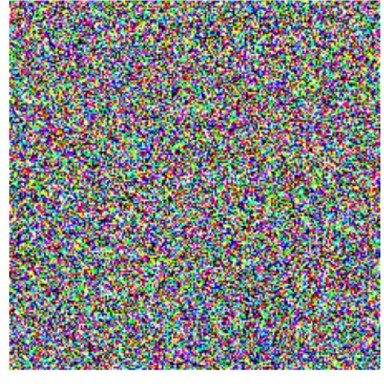

Depth Noise Standard
Deviation: 2.0

Selected Layers:
[1 0 0 0 1 0 0 0 0 0 0 0]

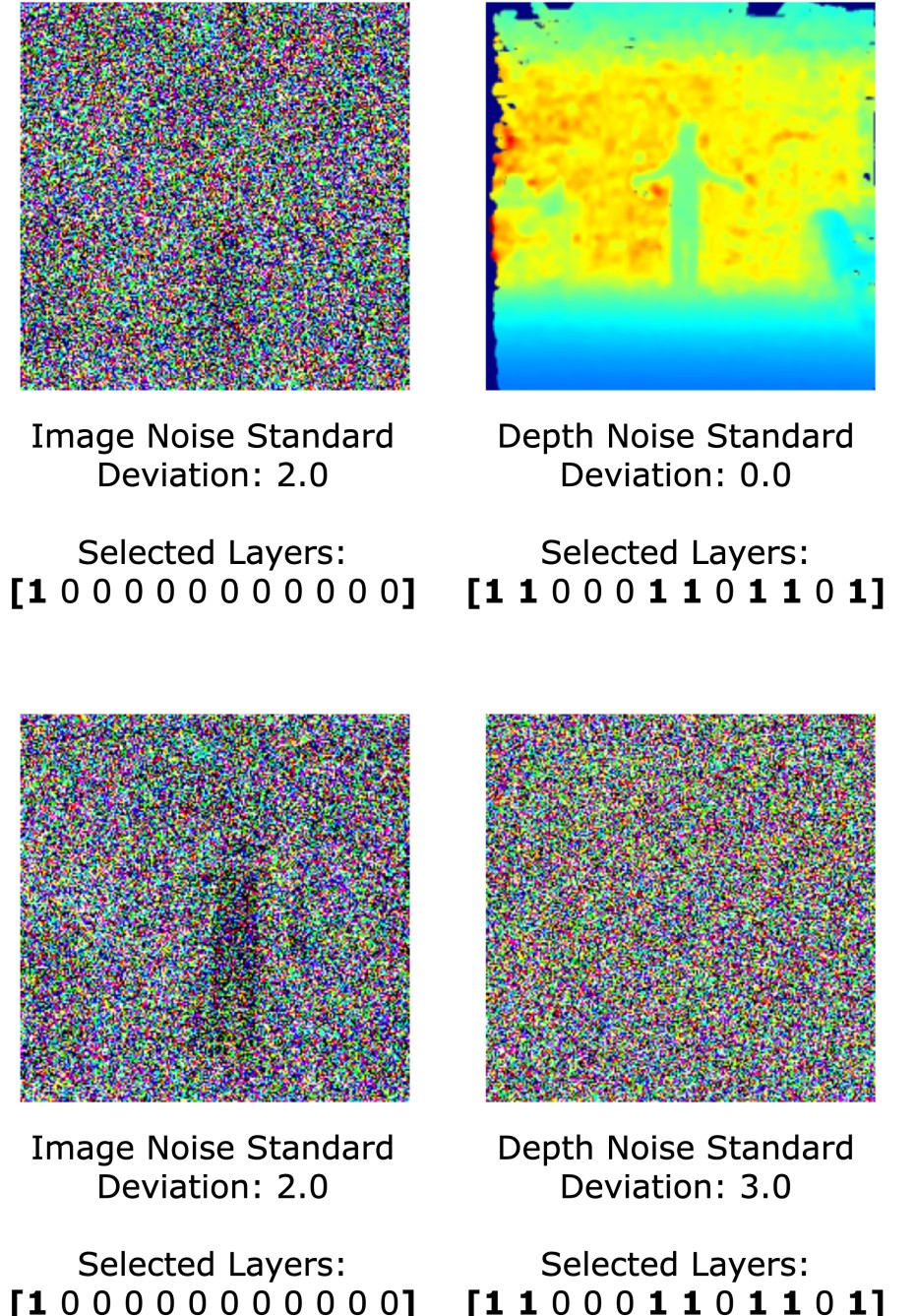

Figure 14: Visual Results on the MM-Fi Dataset highlighting the impact of Gaussian Noise. We showcase the ADMN controller layer allocation

