# OpenReview forum: "ADMN: A Layer-Wise Adaptive Multimodal Network for Dynamic Input Noise and Compute Resources"
_NeurIPS.cc/2025/Conference — NeurIPS 2025 poster_

### Official Review · Reviewer_1X5m · 2025-07-02

**Clarity:** 3
**Significance:** 3
**Originality:** 3
**Rating:** 5
**Confidence:** 3

**Summary:**

This paper proposes a layer wise adaptive multimodal network that has two innovations: 1. the proposed method is able to work within the strict bound of varying compute budgets, where most existing dynamic methods on optimize for average case efficiency. 2. the proposed method is able to adaptively assign compute resource to different input modality, based on their quality of information (QoI), such that the input modality with less corruption is assigned more resources, to achieve optimal performance. The experimental results show that the proposed method indeed achieve better computation consumption / performance tradeoff comparing to a series of non-adaptive baselines, positively supporting the claimed strengths.

**Questions:**

1. Will the proposed controller show the same competitive performance against the baselines when the depth of feature backbones are increased?  since more complex backbones require more sophisticated decision mechanism, I assume this means that the capacity of the controller will need to be increased as well in order to make well informed decisions. is it possible to show, e.g., when the feature backbones are 16 or 20 layers, how many more layers in controller are needed for optimal performance, and whether the increased controller load is still acceptable throughput / FLOP wise.

**Ethical Concerns:**

["NO or VERY MINOR ethics concerns only"]

**Final Justification:**

The author's results regarding the scalability of the proposed controller across multiple modalities is equivalent to increasing the neural network's depth, thus addressing my question. So I will maintain my current rating as accept.

**Limitations:**

Yes

**Paper Formatting Concerns:**

No major issues

**Quality:**

3

**Strengths And Weaknesses:**

Strengths:
1. The paper is well motivated. As in resource limited platforms, optimizing the model within the bound of strict computation budget is more important than optimizing for average case efficiency; Also, modality-quality aware compute allocation is also important to ensure the information among input modalities are reliably utilized, and is a less explored research direction in dynamic network design domain.

2. The method proposed in the paper makes intuitive sense. Firstly, the method finetune the model with layerdrop regularization, ensuring that the network performance will not be affected by layer skipping operation; Further, in the controller training stage, to make sure that the control module is aware of the QoI of the input modality when making compute allocation decision, the paper proposes to add corruption aware prediction loss in supervised setting, or auto-encoding like reconstruction loss in unsupervised setting. And the ablation results show that the added losses indeed facilitate the controlled to make more QoI informed layer allocation predictions.

3. The experimental results show that under the same computation budget, the proposed method achieve better performance than non-adaptive baselines and baselines that are trained from scratch, it also shows that the added computation load of the controller still guarantees high throughput and negligible FLOPS increase. Together the results show that the proposed controller can distribute the compute resource among modalities optimally with acceptable increased model complexity.

4.  The limitations of the current version of the proposed method is clearly discussed.

Weaknesses:
1. The experiment only tests the proposed method when the feature backbones have 12 layers, so it's not clear whether the controller can still work well with increased backbone depth, or how much additional capacity are needed in the controller in order to learn budget allocation with deeper backbones.

---

> ### Author Rebuttal · Authors · 2025-07-30
>
> Dear Reviewer 1x5m,
>
> Thank you so much for your constructive comments, and we look forward to improving our work with your ideas. We are glad to hear that you found the work to be well motivated and the method to be intuitive. We address some key points and questions below:
>
>
> **Point 1: Scaling ADMN to deeper backbones**:
>
> While ADMN’s method is certainly compatible with backbones of any size, we are unable to provide evaluations on larger backbones in the short timespan of the rebuttal due to the training time required. Nevertheless, we are confident that the controller is able to easily scale to larger backbones. The bulk of the controller parameters are devoted to perceiving the modality QoI (e.g., the convolution layers and transformer fusion encoder). The required capacity of these components are related to the difficulty of perceiving the corruption information and are agnostic to the size of the backbone. The only component dependent on the backbone depth is the neural component tasked with translating the corruption embedding into an allocation of layers among backbones. This takes the form of a lightweight, two-layer MLP that projects the corruption embedding into the number of total backbone layers.
>
> In Section 4.3, we showcase ADMN’s ability to handle three modalities, where the controller is capable of accurately allocating resources among 36 layers (12 layers in each of the 3 transformer backbones). We do not alter the controller's architecture, other than changing the output dimension of the MLP from 24 to 36. Thus, while we are unable to directly experiment on larger backbones, this instills confidence that the controller can support significantly more complex backbones (e.g., two backbones each with 18 layers) while maintaining its accuracy and lightweight nature.

---

> > ### Author Response · Authors · 2025-08-05
> > **Rebuttal Follow Up**
> >
> > Dear Reviewer 1x5m,
> >
> > As the rebuttal period is nearing the end, please let us know if you have any additional questions, or if our rebuttal has successfully addressed your concerns. We greatly appreciate your help in improving our work.
> >
> > Thank you so much,
> >
> > Authors

---

> > > ### Comment · Reviewer_1X5m · 2025-08-05
> > >
> > > Thanks for the response, the answer makes sense and addressed my question, thus I will maintain my score

---

### Official Review · Reviewer_dare · 2025-07-03

**Clarity:** 3
**Significance:** 2
**Originality:** 3
**Rating:** 5
**Confidence:** 3

**Summary:**

The paper proposes a way to automatically adapt computation by selecting layers  automatically depending on input quality and computational budget, in a multimodal setting. The main contribution is the use a controller that selects, for each modality, which layers to use. The distribution of layers to choose is automatically learned end-to-end by minimizing the target task loss, and depends on the input data and how much informative is the input, for instance an input which is distorted (e.g. with gaussian blur) will use fewer computation than an an input which is clean.

**Questions:**

Please refer to Strengths And Weaknesses

**Ethical Concerns:**

["NO or VERY MINOR ethics concerns only"]

**Final Justification:**

Explanations given  about attempts to train a controller in a fully end-to-end manner and cause failures were convincing as differentiable selection is a hard problem by itself, and their approach based on autoencoders makes it possible to avoid explicit supervision of corruption level.

**Limitations:**

Yes

**Paper Formatting Concerns:**

No major  formatting issues

**Quality:**

3

**Strengths And Weaknesses:**

The paper is well written and well motivated, dynamic computation in the presence of inputs of varying  quality within strict and constrained resources is an important topic to study. The controller and thus layer selection is trained end-to-end to minimize task loss, which is quite elegant.

Table 5 show that removing the corruption supervision make the results worse. Since the layer selection is trained end-to-end to minimize  the task loss, why can’t the controller figure out automatically to not leverage computation on e.g. severely distorted samples ? could it be related simply the controller network capacity?

The choice of baseline of models is reasonable. However, the Naive Scratch and MNS baselines
are trained from scratch, while ADMN uses pre-trained networks. Could it be that the difference in performance come from the use of pre-trained models ?

Although the controllers are lightweight, it seems redundant to train a controller for each single layer budget. Would it be feasible to have a single controller that generalizes across budgets ?

---

> ### Author Rebuttal · Authors · 2025-07-30
>
> Dear Reviewer dare,
>
> Thank you so much for your constructive comments, and we look forward to improving our work with your ideas. We are glad to hear that you found the work to be well motivated and important to study. We address some key points and questions below:
>
> **Point 1: Inability of the controller to automatically learn the corruption information**:
>
> The obstacle hindering the controller from learning the corruption distribution without any corruption supervision (explicit with metadata or through autoencoder) is the complexity of the training process. Since the selection of a layer is not a differentiable operation, we model it with Gumbel-Softmax Sampling, followed by discretization and a straight through estimator. The gradients received by the controller are thus only an estimation of how that particular layer impacts the downstream loss, with additional complexity arising from the dependence on other layers that were selected alongside it. Consequently, despite training the layer selection mechanism end-to-end, it is very difficult for the earlier perceptual components of the controller to learn to attend to the input modality QoI from the noisy layer gradient information, usually requiring assistance in the form of our corruption-aware supervision or autoencoder initialization.
>
> **Point 2: MNS and Naive Scratch baselines do not utilize pretrained weights**:
>
> The ability to use pretrained weights is a direct advantage of our design choices of ADMN. By specifically using the LayerDrop technique, we can initialize our model with pretrained weights and selectively drop certain layers according to the input modality QoI and the current resource budget. In contrast, the MNS and Naive Scratch methods are not compatible with pretrained weight initialization since pretrained weights only exist for a few configurations of model size and thus must be trained from scratch. We acknowledge that part of ADMN’s performance can be attributed to the usage of pretrained weights, but the drastic benefits over the Naive Alloc baseline, which uses the same weights as ADMN, highlights the importance of intelligent resource allocation. Additionally, the MNS and Naive Scratch baselines have the advantage of training the network for that particular resource budget, while ADMN is forced to use one set of backbone weights for all budgets, which can slightly offset ADMN’s advantage with pretrained weights.
>
>
> **Point 3: Training a Universal Controller**
>
> Thank you for the suggestion! This is a good idea to further extend the generalizability of ADMN by training a universal controller for all layer budgets. To achieve this, we can add an additional budget token representing the number of available layers to the input, or condition the network weights with a hypernetwork on the number of available layers. However, it is possible that the added complexity may result in unstable controller training. We will explore adding this in the camera-ready version of the paper.

---

> > ### Author Response · Authors · 2025-08-05
> > **Rebuttal Follow Up**
> >
> > Dear Reviewer dare,
> >
> > As the rebuttal period is nearing the end, please let us know if you have any additional questions, or if our rebuttal has successfully addressed your concerns. We greatly appreciate your help in improving our work.
> >
> > Thank you so much,
> >
> > Authors

---

> > ### Comment · Reviewer_dare · 2025-08-06
> >
> > Many thanks to the authors for the detailed answer.
> >
> > Regarding Point 1, what would make learning without corruption supervision easier,  did you experiment with
> > e.g. larger controller networks ? or with other ways of doing differentiable layer selection?
> >
> > Regarding Point 2, thanks for clarification. Did you also measure performance of other (naive) allocation baselines ? e.g. randomly assigning layers to each modality.
> >
> > I think moving to fully end-to-end controller learning without corruption supervision and with flexible layer budgets would strengthen the approach, but I understand it may be better suited for future work.

---

> > > ### Author Response · Authors · 2025-08-06
> > >
> > > Dear Reviewer dare,
> > >
> > > Thank you so much for the response!
> > >
> > > **Point 1: End-to-end controller supervision**:
> > >
> > > During our preliminary experiments at the start of this project, we experimented with several variants of the controller network in an attempt to realize fully end-to-end training, from increasing the size of the perceptual components (e.g., more convolutional layers) or increasing the MLP size for layer selection. Unfortunately, we found that the inability for the controller to attend to the QoI was due to the noisy gradients, and cannot be addressed by increasing controller capacity. This is because the gradients backpropagated from the task loss do not provide strong supervision for the perceptual components to attend to modality QoI, especially if the differences in QoI are subtle, which is an architecture agnostic problem.
> > >
> > > In order to train a model despite the non-differentiable layer selection mechanism, we had primarily two methods used in existing literature from other fields - gradient approximation, and reinforcement learning. We employed gradient approximation techniques with a mix of Gumbel-Softmax Sampling and the Straight-Through Estimator. In Appendix A.2, we provide an experiment in which we attempt to use only the Straight-Through Estimator, with poor results. It is possible to employ reinforcement learning, but the added complexity of the approach, with potential problems with convergence, lead us further away from a simple end-to-end approach. We are not aware of other methods to employ that can lead to elevated end-to-end training performance with no corruption supervision.
> > >
> > >
> > > **Point 2: Other Naive Baselines**:
> > >
> > > In our evaluations, aside from Naive Scratch., we explored other Naive baselines including “Naive Allocation”, “Modality 1 Only”, and “Modality 2 Only”. For “Naive Allocation”, we allocate an equal number of layers to each modality, while for “Modality X Only” we allocate all resources to a given modality. We find that ADMN outperforms all these Naive Baselines in Tables 1, 2, 3. We did not explore random selection of layers among the backbones, as it will underperform the Naive Allocation baseline. From the original LayerDrop work, they proposed an “Every Other” dropout strategy, demonstrating that for a given budget, the selection of which layers to drop is important. We follow this strategy for the Naive Allocation baseline. *Randomly selecting layers to activate in a given modality backbone will not adhere to this strategy and will likely result in poor performance.*
> > >
> > > **Point 3: End-to-End with Flexible Layer Budget**:
> > >
> > > We concur that showing fully end-to-end controller training would be a simpler approach. *However, we emphasize that our current autoencoder-based approach can function similarly to fully end-to-end training.* The unsupervised autoencoder approach learns the corruption through the reconstruction objective, and thus does not require any external noise metadata. Given that the autoencoder is a small network (with only a few convolutions and transposed convolutions), performing this training is very fast. *We believe that the low-complexity of this approach, coupled with the fact that we do not require any external noise metadata (same exact dataset as full end-to-end training), makes this approach equally as viable as end-to-end training.*
> > >
> > > In regards to training one controller for all layer budgets, we agree that this would be an effective contribution. However, as we show in Appendix A.1, the training of individual controllers is lightweight and does not constitute a significant bottleneck to training ADMN. Due to the ease of training several dedicated controllers and their small size, if the model architect only wishes to target a small number of layer budgets, it may be simpler to train individual controllers rather than deal with the added complexity of training a universal controller. We will make sure to provide evaluations on universal controllers in the camera-ready version.
> > >
> > > Thank you so much again for the feedback, and please let us know if there are any other questions!

---

> > > > ### Comment · Reviewer_dare · 2025-08-07
> > > >
> > > > Thanks a lot for the detailed answer. I think it would be useful for the reader to include explanations like in Point 1 (above) about your attempts to train a fully end-to-end controller  and issues you encountered. After reading rebuttals and discussion, my concerns were addressed. Therefore, I will increase the score.

---

### Official Review · Reviewer_bi2r · 2025-07-03

**Clarity:** 2
**Significance:** 4
**Originality:** 3
**Rating:** 4
**Confidence:** 3

**Summary:**

This paper introduces ADMN, a layer-wise Adaptive Depth Multimodal Network designed for dynamic environments with fluctuating compute resources and input quality. ADMN adjusts the number of active layers across modalities based on resource constraints and reallocates computation based on modality quality. It achieves similar accuracy to state-of-the-art models while reducing computational cost by up to 75%.

**Questions:**

- Including the main conceptual differences from Mixture-of-Experts approach would be helpful.
- Paper would benefit from backing the motivation behind autoencoder-based initialization (lines 211 - 213) with some evidence.
- In stage 2 of controller training, why is the backbone frozen?
- Minor: Typo in line 2? ‘Afforded’ -> ‘offered’?

**Ethical Concerns:**

["NO or VERY MINOR ethics concerns only"]

**Final Justification:**

After reading the rebuttal and other reviewers’ comments, I am revising my rating of the paper.

The authors have provided clarifications that helped better understand the novelty of the work, particularly the focus on modality-aware, fine-grained resource allocation under strictly bounded compute budgets, which sets it apart from prior dynamic methods. The explanation around the controller design (including both supervised and unsupervised approaches) and its grounding in modality quality adds further depth to the contribution.

Overall, the authors' clarifications have addressed my major concerns, and I now find the paper's contribution more substantial than initially perceived. I am therefore increasing my rating.

**Limitations:**

yes

**Paper Formatting Concerns:**

No concerns.

**Quality:**

3

**Strengths And Weaknesses:**

- The exact novel contribution is not clear. LayerDrop finetuing is based on the existing LayerDrop method, while more clarity on autoencoder-based initialization in QoI-aware training would be helpful.
- Novelty is rather limited.
- Acknowledging this existing work would benefit the paper: Cai, Han, et al. "Once-for-all: Train one network and specialize it for efficient deployment." arXiv preprint arXiv:1908.09791 (2019).
- The topic of the paper is quite relevant from the point of view of practical deployment on edge devices.
- Limitations have been discussed well.

---

> ### Author Rebuttal · Authors · 2025-07-30
>
> Dear Reviewer bi2r,
>
> Thank you so much for your constructive comments, and we look forward to improving our work with your ideas. We address some key points and questions below:
>
> **Point 1: The exact novel contribution is not clear/Novelty is rather limited**
>
> While we acknowledge that ADMN leverages previous techniques such as LayerDrop to enable dynamic layer selection or Gumbel-Softmax Sampling for controller training, ADMN provides several novel contributions both in the novelty of the problem explored and the techniques utilized.
>
> **First, ADMN is the first work to address the impact of relative modality QoI on resource allocation across modalities in multimodal networks, demonstrating that proper allocation can achieve competitive performance with a fraction of the compute.** Previous dynamic techniques such as Early-Exiting reduce unimodal network computation by saving resources on simple inputs and indexing heavily into complex samples. While effective, it is nontrivial to extend these techniques to multimodal networks, especially in realistic scenarios where relative QoIs among modalities vary across time. Additionally, another important factor overlooked by existing dynamic networks is the need to accommodate strictly bounded compute resource budgets. The majority of dynamic networks optimize to lower the *average* resource usage, where difficult samples still fully propagate through the whole network. In contrast, ADMN explores the novel problem of fine-grained resource allocation across modalities while considering a temporally variable but strictly bounded budget.
>
> **Next, ADMN introduces a unique design explicitly structured around resource allocation with respect to variable modality QoI in the presence of fixed resource budgets, proposing two novel methods of grounding a controller network to modality QoI.** Our corruption-aware supervision leverages corruption metadata to train the perceptual components of the controller to attend to modality QoI. In situations where corruption metadata is unavailable, ADMN leverages an unsupervised autoencoder approach encoding modality QoI characteristics into a structured latent space. Previous works have not considered training controllers in the presence of corrupted multimodal samples, much less when the corruption distribution is unknown. Moreover, in Section 3.4.3, ADMN also provides original insights on how to easily extend Gumbel-Softmax Sampling to accommodate selection of a fixed number of layers.
>
> **Finally, while ADMN utilizes the LayerDrop method, we also introduce several innovations on the original work.** The scope of LayerDrop involved only unimodal textual transformers, and ADMN showcases its feasibility on Vision and Audio Transformers through integration with Masked Autoencoder pretraining. Furthermore, ADMN also performs LayerDrop finetuning on multimodal embedding-level fusion networks, and introduces full-backbone dropout to simulate scenarios in which a modality provides little utility.
>
> In summary, ADMN offers novelty in the problem space by exploring the new challenges of resource allocation among corrupted multimodal samples, while simultaneously offering novel technical contributions in QoI-aware controller training and accommodation to fixed resource budgets.
>
>
> **Point 2: Acknowledge existing work “Once-for-all”**
>
> Thank you for sharing this work! We will definitely include it in the related work of our paper. “Once-for-all” provides insights on how an existing large network can be easily pruned down to fit various hardware and energy constraints, offering an interesting alternative to the LayerDrop technique. However, we note that “Once-for-all” does not address the allocation of resources among modalities, which is the primary contribution of ADMN.
>
>
> **Point 3: Differentiating ADMN from Mixture of Experts (MoE)**
>
> We will provide additional clarification in the paper about ADMN’s advantages over MoE techniques. The key differences between ADMN and MoE are (1) MoE's need to train a new model for each compute budget, and (2) its inability to utilize pretrained weights. First, MoE adapts to new compute budgets by training several backbones of various sizes, which incurs a high cost during training. In order to perform fine-grained allocation with MoE, we must train a new model for each allocation. To implement the MNS baseline in Tables 1, 2, 3, we had to train 4 backbones (one for each layer budget) per modality, which was very costly in terms of training resources. In contrast, ADMN requires only the training of one single backbone network with LayerDrop, scaling much more effectively with the number of budgets.
>
> Second, the inability to utilize pretrained weights degrades the MoE accuracy. Typically, loading pretrained weights prior to fine-tuning greatly improves network performance by leveraging strong priors learned from pretraining. Unfortunately, since MoE trains static networks of different sizes, it is likely that pretrained weights do not exist for that particular network size, resulting in lower accuracy when training from scratch, or additional compute-heavy techniques such as knowledge distillation. Conversely, ADMN trains one large network from pretrained weights and dynamically adapts it to runtime conditions.
>
>
> **Point 4: Provide Additional Evidence for Autoencoder-Based Method**
>
> Autoencoders have been known to cluster similar samples close together in the latent space, enabling interpretability in VAEs [1]  or usage in anomaly detection [2]. Based on this insight, we postulate that the autoencoder can learn to encode the corruption information solely through the reconstruction objective without any corruption metadata. This is substantiated by the competitive performance of ADMN_AE, which illustrates that the unsupervised autoencoder training has learned to capture noise characteristics. We will add this additional justification to the paper, along with TSNE plots to visually demonstrate clustering of similar samples in the autoencoder’s latent space. We are unfortunately unable to provide these plots during the rebuttal, as we are limited to text-only responses.
>
>
> **Point 5: Backbone Frozen during Stage 2 Training**
>
> We freeze the backbone during Stage 2 Training for two reasons. First, during Stage 1 Training, the backbone is already exposed to countless configurations of backbone layers as a consequence of employing LayerDrop, and has learned to perform the downstream multimodal task in the presence of missing layers. Thus, unfreezing the backbone in Stage 2 Training will result in the learning of minimal new information. Second, since we may repeat Stage 2 Training for several controllers of different layer budgets, we wish to reduce the number of learnable parameters in the Stage 2 network to expedite training time.
>
>
> **Point 6: “Afforded” vs “Offered”**:
>
> While we used the word “afforded” to indicate “to provide or supply”, we acknowledge that “offered” is a better choice of words.
>
> [1] Luhman, T., & Luhman, E. (2023). High fidelity image synthesis with deep vaes in latent space. arXiv preprint arXiv:2303.13714.
>
> [2] Zimmerer, D., Kohl, S. A., Petersen, J., Isensee, F., & Maier-Hein, K. H. (2018). Context-encoding variational autoencoder for unsupervised anomaly detection. arXiv preprint arXiv:1812.05941.

---

> > ### Author Response · Authors · 2025-08-05
> > **Rebuttal Follow Up**
> >
> > Dear Reviewer bi2r,
> >
> > As the rebuttal period is nearing the end, please let us know if you have any additional questions, or if our rebuttal has successfully addressed your concerns. We greatly appreciate your help in improving our work.
> >
> > Thank you so much,
> >
> > Authors

---

> > ### Comment · Reviewer_bi2r · 2025-08-06
> >
> > I appreciate the detailed clarifications on the points I raised.
> >
> > On novelty, the explanations  made the contribution much clearer. The distinction from early exiting and LayerDrop-based prior work is now well articulated, and I found the controller designs (both supervised and unsupervised) particularly interesting.
> >
> >
> > The clarification on MoE vs ADMN was also very helpful. The comparison around training cost and pretrained weight reuse highlights the practicality of your approach, especially for scenarios where training multiple backbones is infeasible.
> >
> >
> > For the autoencoder-based QoI modeling, the plan to add TSNE visualizations and the justification around clustering in latent space makes sense — I think that will strengthen the interpretability aspect of your approach.
> >
> > And finally, thanks for the clarification on Stage 2 training — freezing the backbone seems reasonable given the prior exposure through LayerDrop.
> >
> > Overall, I think the revisions and added explanations improve the clarity and positioning of the work.

---

> > > ### Author Response · Authors · 2025-08-07
> > >
> > > Dear Reviewer bi2r,
> > >
> > > Thank you so much for your reply! We are glad that we were able to address all of your concerns, and hope that you will consider raising your score.

---

### Note · Authors · 2025-08-13

Dear all,

Thank you so much for your involvement in reviewing this paper, and we are deeply grateful for all the feedback that we have received. We wish to take this opportunity to summarize the key points raised during the rebuttal session.

**Novelty of ADMN**: ADMN is the first work to address the impact of relative modality QoI on resource allocation across modalities in multimodal networks, demonstrating that proper allocation can achieve competitive performance with a fraction of the compute. Besides exploring this novel problem space, we also introduce numerous additional contributions in our QoI-guided controller, fixed-budget gradient propagation, and multimodal extension of LayerDrop.

**Comparison with Naive Scratch and MoE Baselines**: One significant advantage of ADMN over these baselines is its ability to leverage *one single set of network weights* to accommodate all possible resource budgets. Simultaneously, this also enables the usage of pretrained weights, whereas the other architectures that require training for each compute budget cannot. ADMN outperforms these methods both in efficient training (less compute required) and higher accuracy.

**Corruption Aware Training of the ADMN Controller**: We discover that end-to-end training of the ADMN controller fails due to noisy gradients. We propose two methods of addressing this problem - with corruption supervision when metadata is available, and autoencoder initialization when it is not. The organized latent space of the controller allows for learning the corruption distribution without any labels.


During the rebuttal process, we believe that we have successfully addressed the reviewer concerns, and hope that the AC will consider our paper for acceptance.

Best,
Authors

---

### Decision · Program_Chairs · 2025-09-17

**Decision:**

Accept (poster)

**Comment:**

The paper received unanimous acceptance recommendations from the reviewers. The pros include relevance for deployment on edge devices, good writing and elegant design, reasonable choice of baselines, and strong results. Additionally, many reviewers have mentioned that limitations are very well-characterised, which I have validated and agree with - I am mentioning this explicitly, because so many papers pretend to do that accurately but in fact are very one-sided and focus on highlighting only the benefits of their work - the submitted paper is an exception in that regard, and deserves additional praise in my opinion.

At first, the reviewers were not sure about: 1) the novelty of the proposed method, 2) its generalisability, and 3) if the comparison to the baselines is fair. All of those shortcomings were addressed by the authors during the rebuttal, together with providing additional insights into their method. Overall, I find the state of the paper and the associated discussion sufficient for acceptance.